# DOPPLER: DUAL-POLICY LEARNING FOR DEVICE ASSIGNMENT IN ASYNCHRONOUS DATAFLOW GRAPHS

**Xinyu Yao**[1,*]  **Daniel Bourgeois**[1]  **Abhinav Jain**[1]  **Yuxin Tang**[1]  **Jiawen Yao**[1]
**Zhimin Ding**[1]  **Arlei Silva**[1,2]  **Christopher Jermaine**[1]
[1]Rice University  [2]Rice Ken Kennedy Institute
{xy38, dcb10, aj70, yt33, jy75, zd21, arlei, cmj4}@rice.edu

## ABSTRACT

We study the problem of assigning operations in a dataflow graph to devices to minimize execution time in a work-conserving system, with emphasis on complex machine learning workloads. Prior learning-based approaches face three limitations: (1) reliance on bulk-synchronous frameworks that under-utilize devices, (2) learning a single placement policy without modeling the system dynamics, and (3) depending solely on reinforcement learning during pre-training while ignoring optimization during deployment. We propose DOPPLER, a three-stage framework with two policies—SEL for selecting operations and PLC for placing them on devices. DOPPLER consistently outperforms baselines by reducing execution time and improving sampling efficiency through faster per-episode training. Our results show that DOPPLER achieves up to 52.7% lower execution times than the best baseline. The code is available at `https://github.com/xinyuyao/Doppler`.

## 1 INTRODUCTION

Existing systems for multi-GPU computing such as PyTorch (Paszke et al., 2019), TensorFlow (Abadi et al., 2016), and the JAX-based Google stack (Frostig et al., 2018) proceed through a computation in a lock-step, level-wise fashion. Consider the three-matrix multiplication chain $X \times Y \times Z$ in Fig. 1a, each matrix is partitioned into four submatrices to be distributed to eight GPUs to be implemented using the optimal 3D algorithm (Agarwal et al., 1995). The resulting additions, multiplications, and transfers form the dataflow graph in Fig. 1b.

In this graph, $X \times Y$ is first computed through pairwise matrix multiplications (e.g., $X_{11} \times Y_{11}$) distributed across the GPUs. Once these partial results are produced, they must be aggregated and moved, typically using a collective communication library (Woolley, 2015). Then, $Z$ is computed from $X \times Y$, which requires another set of pairwise multiplies, followed by an aggregation step.

In most deep learning systems, such steps are executed bulk-synchronously (Valiant, 1990). For example, until all pairwise multiplications are completed, aggregation cannot begin. This leads to idle resources and lost opportunities for speedup: one slow multiplication delays the entire step (Li et al., 2020c). Moreover, the aggregation phase is communication-dominated, during which GPUs are severely underutilized.

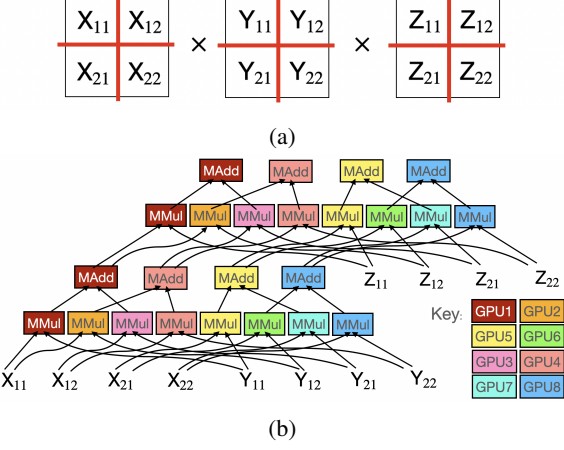

(a)

(b)

Figure 1: A dataflow graph (b) corresponding to the decomposed chain from (a). Vertices correspond to computation kernel calls, edges data dependencies, and colors the mapping of computations to GPUs.

---

*Corresponding Author.

A more efficient approach would overlap "reduce" with computation by scheduling operations (data transfers and kernel calls) asynchronously, as soon as they can be run. A scheduler that never willingly allows resources to sit idle and schedules them dynamically is called *work-conserving* (WC) (Kleinrock, 1965). Table 1 shows the potential speedup of a WC system on two workloads: a chain of matrix multiplications and additions, CHAINMM, and a feedforward neural network, FFNN (configuration details are provided in Appendix F). On a GPU server, the WC system achieves a reduction of 46.3 ms (33%) for CHAINMM and 26.7 ms (53%) for FFNN, compared to PyTorch. In the context of long-term deployment, these small per-query millisecond reductions accumulate into substantial GPU-hour savings. For instance, a 24.2 ms reduction per query for running a Llama model (Yaadav, 2024) amounts to more than 2.4 million GPU hours saved annually at ChatGPT-scale workloads (assuming one million queries per day). Additional results are shown in Section 6.

While WC systems can improve efficiency, they introduce significant challenges, particularly for GPU assignment. In contrast to bulk-synchronous settings that fix execution order via all-reduce, WC systems rely on asynchronous point-to-point communication; without global synchronization, execution order is harder to control, and performance becomes highly sensitive to hardware heterogeneity and resource

| MODEL | WC SYSTEM | SYNCHRONOUS |
|---|---|---|
| CHAINMM | 139 | 185.3 |
| FFNN | 50.2 | 76.9 |

Table 1: Execution time (in milliseconds) for execution in a work-conserving system and a synchronous system.

contention. An effective device assignment must capture this information and balance two competing goals: (1) maintaining GPU load balance and (2) minimizing inter-GPU communication (Harlap et al., 2018; Lu et al., 2017). Traditional placement methods emphasize communication minimization, but under a WC scheduler, the stochastic execution order makes load balancing especially challenging, as **load balancing is inherently temporal** (Saha et al., 2019).

In this paper, we tackle the device assignment problem in an asynchronous WC system by introducing DOPPLER, a reinforcement learning–based framework that adopts a learning-by-doing paradigm to optimize device placement. While prior work Addanki et al. (2019); Zhou et al. (2019); Mirhoseini et al. (2017) learns a single placement policy, DOPPLER learns efficient kernel assignments through a dual-policy sequential decision scheme. The first policy selects the next kernel (vertex in the dataflow graph) to assign by traversing the partially assigned dataflow graph in a manner that approximates the non-deterministic flow of "time," while the second policy determines the GPU placement of that kernel to balance load and minimize communication. This separation captures both execution dynamics and hardware constraints, producing assignments tailored to stochastic WC execution.

Moreover, DOPPLER employs a three-stage framework for training its dual policies. Stage I (offline) uses supervised learning to train the policies to follow simple heuristics, such as co-locating neighboring vertices on the same device. Stage II (offline) transitions to reinforcement learning: the policies generate assignments that are "executed" in a simulated WC system, with rewards computed from the simulated runtime. Stage III (online) deploys the trained policies in a real WC system, where they are continuously refined through reinforcement learning using rewards derived from observing assignment runtimes, and the policies are recursively updated as the system executes.

Our contributions are the following: (1) We investigate the device assignment problem in a multi-GPU system under a WC system; (2) We introduce a dual-policy learning approach to first learn the approximated traversal order of nodes before assigning them to devices; (3) We propose DOPPLER, a three-stage training framework to improve scheduling efficiency on the fly by continuously training during deployment, along with two pretraining stages to accelerate convergence during deployment; (4) Our experiments show that DOPPLER achieves up to 52.7% lower execution times compared to the best baseline. DOPPLER also achieves a significant runtime reduction compared to a stronger baseline that we designed (ENUMERATIVEOPTIMIZER), by up to 13.8%.

## 2    DEVICE ASSIGNMENT PROBLEM IN A WORK-CONSERVING SYSTEM

In this paper, we study the device assignment problem for static dataflow graphs, which capture modern ML workloads. State-of-the-art architectures—such as Transformers (Vaswani et al., 2017) and Mixture-of-Experts models (Jacobs et al., 1991; Shazeer et al., 2017)—are defined over largely fixed computation graphs during both training and inference (Shi et al., 2024). Crucially, **while the**

**graph structure is static, its execution is fully asynchronous**: the runtime schedules and executes operators in parallel, constrained only by data dependencies in the graph.

Formally, consider a *dataflow graph* $\mathcal{G} = \langle \mathcal{V}, \mathcal{E} \rangle$, where $\mathcal{V} = \{v_1, v_2, \ldots, v_n\}$ denotes a set of vertices representing computations, and $\mathcal{E} = \{e_1, e_2, \ldots, e_m\}$ denotes directed edges representing data dependencies. Given a set of devices $\mathcal{D}$, our goal is to compute a device assignment $A : \mathcal{V} \to \mathcal{D}$ that maps each vertex $v \in \mathcal{V}$ to a device in $\mathcal{D}$. We denote by $A_v$ the device assigned to vertex $v$ under assignment $A$. The objective is to find an assignment $A$ that minimizes the overall execution time, denoted by ExecTime$(A)$.

For an example dataflow graph, consider the matrix multiplication chain $X \times Y \times Z$, which can be decomposed to run on a server with eight GPUs by sharding each matrix four ways in Fig. 1a. The resulting fine-grained dataflow graph (Fig. 1b) contains eight submatrix multiplies associated with the two original multiplies, and four matrix additions to aggregate the results. A candidate assignment of the graph to eight GPUs is shown in Fig. 1b. This assignment achieves low execution time as (a) the expensive matrix multiplications that tend to run in parallel are load-balanced, and (b) communication is minimized by co-locating neighboring nodes.

---

**Algorithm 1** ExecTime$(A)$

---

$\%$ $rdy[v, d]$ is T iff the result of vertex
$\%$ $v$ is on device $d$; initially, nothing is ready
$rdy[v, d] \leftarrow$ F $\ \forall (v \in \mathcal{V}, d \in \mathcal{D})$
$\%$ except inputs: available everywhere
$rdy[v, d] \leftarrow$ T $\ \forall (v \in \mathcal{V}, d \in \mathcal{D})$ s.t. $(v', v) \notin \mathcal{E}$
$t \leftarrow 0$ $\%$ exec begins at time 0
$S \leftarrow \langle \rangle$ $\%$ schedule is empty
**while** $\exists (v \in \mathcal{V})$ s.t. $rdy[v, A_v] =$ F **do**
$\quad tasks \leftarrow$ EnumTasks$(rdy, A, S)$
$\quad task \leftarrow$ ChooseTask$(rdy, A, S, tasks)$
$\quad$**if** $task =$ null **then**
$\quad\quad \%$ if no task is chosen, just wait
$\quad\quad \langle t, task \rangle \sim P(.|S, t)$ $\%$ which $task$ done?
$\quad\quad S \leftarrow S + \langle task, t, \text{end} \rangle$ $\%$ save completion
$\quad\quad rdy[\text{vertex}(task), \text{device}(task)] \leftarrow$ T
$\quad$**else**
$\quad\quad S \leftarrow S + \langle task, t, \text{beg} \rangle$ $\%$ record initiation
$\quad$**end if**
**end while**
**return** $t$

---

Therefore, a key question is **how to define the execution time of an assignment ExecTime$(A)$ in WC systems?** In practice, it is difficult to derive a closed formula for ExecTime$(A)$, given the stochasticity of WC systems. As operations are issued dynamically, based on the state of the system, different runs of the same assignment can have very different execution times.

Algorithm 1 describes how an assignment $A$ is executed in a WC system (the subroutine EnumTasks$(rdy, A, S)$ in Algorithm 2 enumerates the tasks that can be taken at each step by the scheduler). The algorithm stochastically simulates the execution of the assignment $A$ via a WC dynamic scheduler and returns the total execution time. It works by repeatedly asking the scheduler to choose the next task to schedule; when there is no task that can be

---

**Algorithm 2** EnumTasks$(rdy, A, S)$

---

$output \leftarrow \{\}$
$\%$ get all potential transfers
**for** $(v_1, v_2) \in \mathcal{E}$ **do**
$\quad$**if** $rdy[v_1, A_{v_2}] =$ false and $rdy[v_1, A_{v_1}] =$ true and transfer$(v_1, A_{v_1}, A_{v_2}) \notin S$ **then**
$\quad\quad$Add transfer$(v_1, A_{v_1}, A_{v_2})$ to $output$
$\quad$**end if**
**end for**
$\%$ get all potential ops to exec
**for** $v_2 \in \mathcal{V}$ **do**
$\quad$**if** $rdy[v_1, A_{v_2}] =$ true $\forall v_1$ s.t. $(v_1, v_2) \in \mathcal{E}$ and exec$(v_2, A_{v_2}) \notin S$ **then**
$\quad\quad$Add exec$(v_2, A_{v_2})$ to $output$
$\quad$**end if**
**end for**
**return** $output$

---

scheduled, the algorithm waits until an event is stochastically generated. In the algorithm, a *schedule* $S$ is the complete list of events that have occurred up to $t_{in}$. An *event* is a (*task*, *time*, *eventtype*) triple, where *task* is either an data transfer transfer between devices or an execution (exec) of a node on a device, *time* records when the transfer or exec event occurs, and *eventtype* specifies the recorded time as either beg or end of an event. EnumTasks enumerates all transfer and exec tasks that are ready when EnumTasks is called.

Algorithm 1 has two key generic components that allow it to serve as a reasonable proxy, or digital twin, for a real-life scheduler executed on real-life hardware. First, the distribution $P(\langle t_{out}, a \rangle \mid$

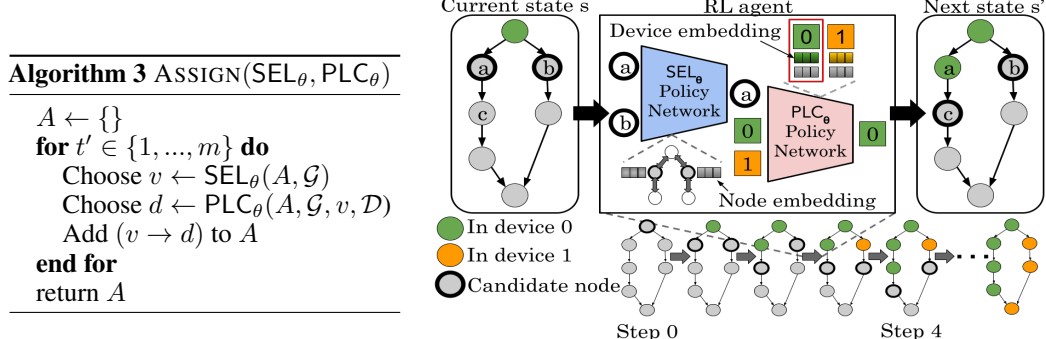

**Algorithm 3** ASSIGN($\mathsf{SEL}_\theta, \mathsf{PLC}_\theta$)

$A \leftarrow \{\}$
**for** $t' \in \{1, ..., m\}$ **do**
    Choose $v \leftarrow \mathsf{SEL}_\theta(A, \mathcal{G})$
    Choose $d \leftarrow \mathsf{PLC}_\theta(A, \mathcal{G}, v, \mathcal{D})$
    Add $(v \rightarrow d)$ to $A$
**end for**
return $A$

Figure 2: (Left) The ASSIGN algorithm, which sequentially produces an assignment $A$ using $\mathsf{SEL}_\theta$ policy and placed using $\mathsf{PLC}_\theta$ policy. (Right) A graphical depiction of the algorithm's implementation.

$S, t_{in}$) governs the "next completed task." Given a schedule $S$ and a current time $t_{in}$, $P$ is a joint distribution over the next task to complete and the time $t_{out}$ at which this task completes. Second, the function `ChooseTask` encapsulates the underlying scheduling algorithm that is implemented by the WC system. It may choose any *task* from a set of *tasks*. As described, it may operate depth-first (seeking to probe deeply into $\mathcal{G}$), breadth-first, or may employ any other applicable strategy.

In practice, Algorithm 1 is implemented by either (a) a simulator, where the distribution $P(\langle t_{out}, a \rangle \mid S, t_{in})$ is realized by a model that takes into account factors such as the number of floating-point operations in the underlying operation (e.g., CUDA kernel executions) or the number of bytes to be transferred (in the case of GPU-to-GPU communication), thereby simulating the event-driven behavior of a real system, or (b) deploying the assignment $A$ in a real-world WC system and observing the runtime. We use option (a) in Stage II of DOPPLER and option (b) in Stage III of DOPPLER (see details in Section 5).

## 3   A SOLUTION VIA REINFORCEMENT LEARNING

As there is no closed-form objective function, learning-by-doing is a promising approach for determining the optimal assignment $A^*$. Stages II and III of DOPPLER formulate choosing the assignment as a bandit problem. For an assignment $A$ at time tick $t$, we obtain a reward $r_t \sim R_A$ where $R_A$ is the reward distribution for $A$ (in practice, $r_t$ is sampled by invoking `ExecTime`($A$) and observing the runtime). Let $R^* = R_{A^*}$, where $A^* = \arg\max_A \mathbb{E}[R_A]$. Our goal is to minimize the regret:

$$\rho = \sum_{t=1}^{T} \left( \mathbb{E}[R^*] - r_t \right) \tag{1}$$

This is a bandit problem with $\mathcal{D}^{|\mathcal{V}|}$ arms. Our problem is not amenable to classic solutions because the number of arms is so large. For example, if we are producing an assignment for an 8-GPU server that is to execute a dataflow graph with 100 vertices, there are $\approx 2^{300}$ possible assignments.

Fortunately, there is a combinatorial structure to the assignment problem that can allow us to deal with the very large set of possible assignments (Cesa-Bianchi & Lugosi, 2012; Chen et al., 2013). Note that if two assignments $A_1$ and $A_2$ differ only in how a few vertices have been assigned to devices, it is likely that the two reward distributions $R_{A_1}$ and $R_{A_2}$ will be similar. Thus, it may be possible to systematically search the possible assignments.

Our approach builds the assignment at each time tick using a sequential process controlled by two policies $\mathsf{SEL}_\theta$ and $\mathsf{PLC}_\theta$. If we use ASSIGN($\mathsf{SEL}_\theta, \mathsf{PLC}_\theta$) (Algorithm 3) as the mechanism for choosing the assignment $A$ at each time tick $t$ of the bandit problem in Equation 1, this process can be reformulated as an episodic Markov decision process (MDP), as shown in Liu et al. (2024), where each episode executes ASSIGN($\mathsf{SEL}_\theta, \mathsf{PLC}_\theta$) (shown in Fig. 2) and a reward is obtained once the assignment is completed at the end of each episode. Therefore, any suitable reinforcement learning algorithm can be used to learn the policies $\mathsf{SEL}_\theta$ and $\mathsf{PLC}_\theta$. In our implementation, we apply graph

neural networks (GNN) along with message passing to encode graph $\mathcal{G}$ and use feedforward neural networks to decode actions for $\mathsf{SEL}_\theta$ and $\mathsf{PLC}_\theta$. Details on GNN architectures are in Section 4.2.

# 4 DOPPLER DUAL POLICY IMPLEMENTATIONS

We describe our episodic Markov Decision Process formulation (Section 4.1), along with the graph neural network architectures that we used to implement the dual policy learned during DOPPLER training (Section 4.2) and an efficiency analysis of the GNN message-passing (Section 4.3).

## 4.1 EPISODIC MDP FORMULATION

We formulate device assignment as an episodic Markov Decision Process $(\mathcal{S}, \mathcal{A}, H, \mathcal{P}, \mathcal{R})$, where $\mathcal{P}$ is the transition function, and $H$ is the horizon, which equals the number of nodes in the graph.

**States** Each state $s_h \in \mathcal{S}$ is a tuple $(X_\mathcal{G}, \mathcal{C}_h, X_{\mathcal{D},h})$, where $X_\mathcal{G} = (X_\mathcal{V}, X_\mathcal{E})$ represents the static graph features, including node and edge features such as bottom-level paths (i.e., to entry nodes), top-level paths (i.e., to exit nodes), and communication costs. $\mathcal{C}_h$ is the dynamic set of candidate nodes, and $\mathcal{C}_0$ is defined as the set of entry nodes in the graph. Finally, $X_{\mathcal{D},h}$ represents the dynamic device features (e.g., total computing time and the end time for computations on each device). Details about the sets of features can be found in Appendix E.

**Actions** At each time step $h$, the agent takes an action $a_h \in \mathcal{A}$, where $a_h = (v_h, d_h)$. Each action selects a node $v_h$ and places it on device $d_h$ using policies $\mathsf{SEL}\theta$ and $\mathsf{PLC}\theta$, as shown in Fig. 2. Each episode is composed of $|\mathcal{V}|$ iterations—i.e., one iteration per node in $\mathcal{V}$.

**Reward** We calculate rewards using the execution time $R_{s_H} = (-1) \times \texttt{ExecTime}(s_H)$ derived from either the real system or a simulator. The reward is computed at the end of each episode $(h = H)$, with intermediate rewards set to zero for efficiency. To enhance stability, we subtract a baseline reward equal to the average execution time observed across all previous episodes $(\overline{R_{s_H}})$. The final reward is computed as $r_H = R_{s_H} - \overline{R_{s_H}}$.

## 4.2 DUAL POLICY GRAPH NEURAL NETWORK ARCHITECTURES

Device assignment requires reasoning over dataflow graphs whose behavior is shaped by dependencies, critical paths, and communication edges. Graph neural networks naturally provide the right inductive bias for this setting: they propagate information along dependency edges, capture local structure, and generalize across graphs of varying sizes. The dual-policy decomposition is motivated by heterogeneity, since decisions depend jointly on both the node being scheduled and the device on which it may run. This modular design reflects the semantics of the dynamic environment and allows the same architecture to adapt across different workloads and hardware configurations.

We further describe the details of the policy networks for computing $\mathsf{SEL}_\theta$ and $\mathsf{PLC}_\theta$ in Algorithm 3. The symbol $\theta$ is used here to emphasize that these functions have parameters that will be optimized as part of the training process. DOPPLER applies a graph neural network (GNN) to encode node information in the dataflow graph. Our GNN is a message-passing neural network (Gilmer et al., 2017) that learns node representations for each node $v$ via $K$ successive iterations:

$$\mathbf{h}_v^{[k]} = \phi(\mathbf{h}_v^{[k-1]}, \bigoplus_{u \in N(v)} \psi(\mathbf{h}_u^{[k-1]}, \mathbf{h}_v^{[k-1]}, \mathbf{e}_{uv})) \tag{2}$$

where $\mathbf{h}_v^{[k]}$ are representations learned at the $k$-th layer, $\mathbf{h}_v^{[0]} = X_\mathcal{V}[v]$, $\psi$ and $\phi$ are functions, $N(v)$ are the neighbors of $v$, and $\bigoplus$ is a permutation-invariant operator. We will use $\mathsf{GNN}(\mathcal{G}, X_\mathcal{G}) = [\mathbf{h}_1^{[K]}; \mathbf{h}_2^{[K]}; \dots; \mathbf{h}_n^{[K]}]$ to refer to the representations of all nodes in $\mathcal{G}$.

We also apply an $L$-layer feedforward neural network (FFNN) to encode node information: $\mathbf{x}_v^{[l]} = W^{[l]}\mathbf{x}_v^{[l-1]} + \mathbf{b}^{[l]}$ where $\mathbf{x}_v^{[l]}$ are representations at the $l$-th layer, and $W^{[l]}$ and $\mathbf{b}^{[l]}$ are weights and biases, respectively. Let $\mathsf{FFNN}(X) = [\mathbf{x}_1^{[L]}; \mathbf{x}_2^{[L]}; \dots \mathbf{x}_n^{[L]}]$ be the representations of all nodes in $\mathcal{G}$, with $\mathbf{x}_v^{[0]} = X[v]$.

**Node policy network** ($\mathsf{SEL}_\theta$)**:** Selects a node from the candidate set $\mathcal{C}$ based on the observed graph state $X_\mathcal{G}$ using the $\epsilon$-greedy approach. Let $b(v)$ and $t(v)$ be the b-path and t-path for $v$, where a b-level (t-level) path for $v$ is the longest path from $v$ to an entry (exit) node in $\mathcal{G}$. We aggregate information from these critical paths via GNN embeddings $H[u]$ for each node $u$ along them. Nodes are selected according to probabilities estimated from a graph embedding matrix $H_\mathcal{G}$, which is the result of the concatenation of critical path ($\mathbf{h}_{v,b}$ and $\mathbf{h}_{v,t}$), GNN ($H[v]$), and feature ($Z[v]$) representations.

$$H = \mathsf{GNN}(\mathcal{G}, X_\mathcal{G}), \quad Z = \mathsf{FFNN}(X_\mathcal{V}), \quad \mathbf{h}_v = \Big[ H[v] \, \| \, \mathbf{h}_{v,b} \, \| \, \mathbf{h}_{v,t} \, \| \, Z[v] \Big], \tag{3}$$

$$Q_\mathcal{G}(v) = \mathsf{softmax}\big(\mathsf{FFNN}(\mathbf{h}_v)\big), \quad \mathsf{SEL}_\theta(\mathcal{G}, X_\mathcal{G}) = \begin{cases} \arg\max_v Q_\mathcal{G}(v), & p = 1 - \epsilon, \\ \text{random } v \in \mathcal{C}, & p = \epsilon. \end{cases} \tag{4}$$

where $p$ is the probability of the event and $\epsilon$ is a parameter.

**Device policy network** ($\mathsf{PLC}_\theta$)**:** Places a node $v$ into one of the devices in $\mathcal{D}$ based on the composed state observation ($v, X_\mathcal{D}, X_\mathcal{G}$). Devices are selected based on an embedding matrix $H_\mathcal{D}$ for the set of devices, generated by concatenating representations for the node ($H[v]$), node features ($Z[v]$), device features ($Y[d]$) and nodes already placed on the device ($\mathbf{h}_d$).

$$H = \mathsf{GNN}(\mathcal{G}, X_\mathcal{G}), \quad Y = \mathsf{FFNN}(X_\mathcal{D}), \quad Z = \mathsf{FFNN}(X_\mathcal{G}), \tag{5}$$

$$\mathbf{h}_{v,d} = \big[ H[v] \, \| \, \mathbf{h}_d \, \| \, Y[d] \, \| \, Z[v] \big], \quad H_\mathcal{D} = \big[ \mathbf{h}_{v,1}; \dots; \mathbf{h}_{v,m} \big], \tag{6}$$

$$H'_\mathcal{D} = \mathsf{LeakyReLU}\big(\mathsf{FFNN}(H_\mathcal{D})\big), \quad Q_\mathcal{D}(d) = \mathsf{softmax}\big(\mathsf{FFNN}(H'_\mathcal{D})\big), \tag{7}$$

$$\mathsf{PLC}_\theta(\cdot) = \begin{cases} \arg\max_d Q_\mathcal{D}(d), & p = 1 - \epsilon, \\ \text{random } d \in \mathcal{D}, & p = \epsilon. \end{cases} \tag{8}$$

### 4.3 Efficient Message Passing Approximation

Implementing the MDP described in Section 4.1 requires performing message-passing on the dataflow graph $\mathcal{G}$ when calling $\mathsf{SEL}_\theta$ and $\mathsf{PLC}_\theta$ policies at each MDP step $h$. We found this to be prohibitive for large graphs since we may apply up to $8k$ episodes $\times$ 261 steps = 2m steps in our experiments. PLACETO (one of our baselines) suffers from this issue, being very inefficient during training as it performs one message-passing round per MDP step. Instead, we propose performing message passing on the graph only once per MDP episode and encoding updated assignment information at each step $h$ in $X_{\mathcal{D},h}$ without message passing. We found empirically that this modification has a negligible impact on DOPPLER's convergence but leads to a significant reduction in training time, especially for large neural networks (we show an ablation study in Appendix G.3).

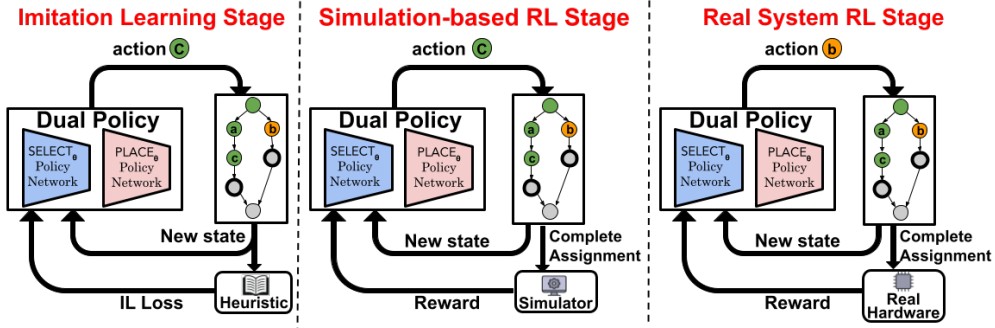

Figure 3: Three-stage framework for cost-effective training of DOPPLER, combining imitation learning, simulation-based reinforcement learning, and real-system reinforcement learning.

## 5 DOPPLER: COST-EFFECTIVE TRAINING

DOPPLER adopts a three-stage framework to train and deploy its dual policies ($\text{SEL}_\theta$ and $\text{PLC}_\theta$): (Stage I) imitation learning, (Stage II) simulation-based reinforcement learning, and (Stage III) real-system reinforcement learning (RL) as shown in Fig. 3. Imitation learning serves as a pre-training step that accelerates convergence in subsequent RL stages by leveraging an existing list-scheduling heuristic. The two RL stages further realize a "learning-by-doing" paradigm, enabling the policies to be continuously refined through interaction with simulated and real systems.

**Imitation Learning stage (Stage I).** We propose using imitation learning for teaching the dual policy to replicate the decisions of an existing heuristic (teacher) before deploying it to the real system. During imitation learning, the computation graph emits the current assignment state at each step, which is fed to both the dual policy and the heuristic. We then apply the decision of the CRITICAL PATH heuristic ($a_{cp}$) to obtain the objective function:

$$J(\theta) = \mathbb{E}_{a_{cp}\sim\Pi_{cp}(s),s\sim T(s',a),a\sim\Pi_\theta(s')}[\nabla_\theta \log \Pi_\theta(a_{cp}|s)] \tag{9}$$

**Simulation-based reinforcement learning stage (Stage II).** Even after pre-training with a teacher, we may have a policy that is too low quality for deployment in the real system, where longer running times or slow convergence due to exploration may be unacceptable. Thus, we also train DOPPLER using a software-based simulator that implements Algorithm 1. The simulator is invoked only at the end of an episode, after we obtain the assignment for all nodes. The dual-policy networks are updated using the policy gradient method (Sutton et al., 1999) by maximizing the following objective function:

$$J(\theta) = \mathbb{E}_{a\sim\Pi_\theta(s)}[\nabla_\theta(\log \Pi_\theta(a|s))R(s,a)] \tag{10}$$

**Real-system reinforcement learning stage (Stage III).** This stage is analogous to sim-to-real adaptation in RL-based hardware scheduling as no simulator can perfectly capture dynamic GPU contention, NVLink jitter, or system-level scheduling noise. Because the first two stages already produce a high-quality dual policy, deployment does not subject users to long warm-up periods or unstable exploratory behavior. Stage III performs lightweight online refinement: once deployed, the policy adapts to the target system's specific hardware configurations and workload characteristics. Importantly, *the reward signal for optimizing Equation 10 is obtained "for free"*: the system naturally observes real execution times (`ExecTime`) while serving requests. This allows DOPPLER to continuously improve in production with no additional overhead or disruption to users.

| MODEL | 4 GPUs | | | | | | RUNTIME REDUCTION | |
| --- | --- | --- | --- | --- | --- | --- | --- | --- |
| | CRIT. PATH | PLACETO | GDP | ENUMOPT. | DOPPLER-SIM | DOPPLER-SYS | BASELINE | ENUMOPT. |
| CHAINMM | $230.4 \pm 4.3$ | $137.1 \pm 2.2$ | $198.0 \pm 3.3$ | $139.0 \pm 10.0$ | $\mathbf{122.5} \pm 4.0$ | $123.4 \pm 2.5$ | 10.7% | 11.9% |
| FFNN | $217.8 \pm 11.3$ | $126.3 \pm 5.8$ | $100.3 \pm 3.2$ | $50.2 \pm 2.5$ | $49.9 \pm 1.1$ | $\underline{\mathbf{47.4}} \pm 0.7$ | 52.7% | 5.6% |
| LLAMA-BLOCK | $230.9 \pm 8.7$ | $411.5 \pm 19.7$ | $336.5 \pm 8.4$ | $172.7 \pm 5.0$ | $191.5 \pm 6.0$ | $\underline{\mathbf{160.3}} \pm 4.3$ | 30.6% | 7.2% |
| LLAMA-LAYER | $292.6 \pm 5.8$ | $295.1 \pm 7.0$ | $231.5 \pm 5.1$ | $174.8 \pm 4.7$ | $167.0 \pm 3.4$ | $\underline{\mathbf{150.6}} \pm 4.2$ | 48.5% | 13.8% |

Table 2: Real engine execution time (ms) for assignments produced by our methods (ENUMOPT., DOPPLER) and prior baselines. We also show the runtime reductions achieved by DOPPLER-SYS compared with previously proposed baselines and ENUMOPT.

## 6 EXPERIMENTS

We study the following questions, focusing on the quality of assignments produced by DOPPLER (Q1–Q5) and the scalability of DOPPLER's dual-policy networks (Q6). Q1: How does DOPPLER compare with alternative approaches in terms of the execution time achieved by its generated assignments? Q2: What are the individual contributions of the $\text{SEL}_\theta$ and $\text{PLC}_\theta$ policies to assignment performance? Q3: How do imitation learning (Stage I), simulation-based RL (Stage II), and real-system RL (Stage III) jointly improve the performance of DOPPLER's training? Q4: How can execution time be explained based on the assignments produced by DOPPLER? Q5: Do DOPPLER's learned dual-policy generalize from one dataflow graph to unseen graphs and hardware architectures? Q6: How do DOPPLER's policy training cost and inference overhead scale with graph size?

## 6.1 EXPERIMENTAL SETUP

**Neural network architectures.** We test dataflow graphs from four types of neural network architectures in our experiments: a feedforward neural network (FFNN), chain matrix multiplications (CHAINMM), a Llama transformer block (LLAMA-BLOCK), and a complete Llama transformer layer (LLAMA-LAYER). FFNN and CHAINMM capture the dense linear-algebra–dominated workloads common across classical neural networks Markidis et al. (2018), while transformer blocks represent the dominant architecture in contemporary large language models Vaswani et al. (2017). Further details can be found in Appendix D.

**GPU systems.** We compare the assignment approaches using four NVIDIA Tesla P100 GPUs with 16GB of memory each. Moreover, we conducted ablation studies on (1) restricted GPU memory (8GB out of 16GB) and (2) eight NVIDIA V100 GPUs with 32GB of memory, and we show the results in Appendix H.

**Baselines.** We compare our approach against four baselines. CRITICAL PATH Kwok & Ahmad (1999) is a popular (non-learning) heuristic for DAG device assignment. PLACETO Addanki et al. (2019) is a recent RL-based alternative that applies a single (device) policy and is trained using simulations (see the ablation study on the simulator in Appendix G.1). GDP Zhou et al. (2019) is another recent RL-based method that consists of graph embedding and sequential attention. ENUMERATIVEOPTIMIZER is a baseline we developed—it is our best effort at exploiting the structure of a sharded tensor computation to produce a high-quality assignment (described in detail in Appendix B).

|  | 4 GPUs | | |
|---|---|---|---|
| MODEL | SYS | SEL | PLC |
| CHAINMM | 123.4±2.5 | 127.0±0.8 | **121.6**±0.7 |
| FFNN | **47.4**±0.7 | 59.1±7.6 | 63.2±1.6 |
| LLAMA-BLOCK | **160.3**±4.3 | 175.6±4.1 | 172.9±4.3 |
| LLAMA-LAYER | **150.6**±4.2 | 161.7±4.1 | 159.5±4.9 |

Table 3: Ablation study showing real engine execution times (ms) on 4 GPUs. SYS, SEL, and PLC denote DOPPLER-SYS, DOPPLER-SEL, and DOPPLER-PLC, respectively.

**Hyperparameters.** For RL-based methods, we run 4k episodes for CHAINMM and FFNN, and 8k episodes for LLAMA-BLOCK and LLAMA-LAYER. We tried different learning rate schedules for each method (initial values $\{1e-3, 1e-4, 1e-5\}$) and found that $1e-3$ decreasing linearly to $1e-6$ works best for PLACETO and $1e-4$ linearly decreasing to $1e-7$ works best for all versions of DOPPLER and GDP. We apply a $0.5$ exploration rate linearly decreasing to $0.0$ for PLACETO and $0.2$ linearly decreasing to $0.0$ for DOPPLER and GDP. An entropy weight of $1e-2$ is applied for all RL methods. For CRITICAL PATH, we run 50 assignments and report the best execution time. The reported execution time (and standard deviation) using the real system is the average of 10 executions.

| | | 4 GPUs | | | | | | |
|---|---|---|---|---|---|---|---|---|
| TRAIN MODEL | TARGET MODEL | ZERO-SHOT | 2K-SHOT | 4K-SHOT | DOPPLER-SYS | CRIT. PATH | PLACETO | ENUMOPT. |
| FFNN | LLAMA-BLOCK | 251.0 ± 2.9 | 165.3 ± 4.1 | **159.4** ± 4.8 | **160.3** ± 4.3 | 230.9 ± 8.7 | 411.5 ± 19.7 | 172.7 ± 5.0 |
| CHAINMM | LLAMA-BLOCK | 242.3 ± 6.7 | 184.9 ± 4.3 | **174.0** ± 4.4 | **160.3** ± 4.3 | 230.9 ± 8.7 | 411.5 ± 19.7 | 172.7 ± 5.0 |
| FFNN | LLAMA-LAYER | 206.1 ± 4.5 | 158.2 ± 4.1 | **155.8** ± 5.0 | **150.6** ± 4.2 | 292.6 ± 5.8 | 295.1 ± 7.0 | 174.8 ± 4.7 |
| CHAINMM | LLAMA-LAYER | 338.2 ± 5.0 | 164.4 ± 3.3 | **156.4** ± 4.4 | **150.6** ± 4.2 | 292.6 ± 5.8 | 295.1 ± 7.0 | 174.8 ± 4.7 |

Table 4: Real engine execution time (ms) for assignments produced by DOPPLER under different few-shot transfer settings (zero-shot, 2k-shot, 4k-shot) compared against full training (DOPPLER-SYS) and baselines. DOPPLER achieves performance comparable to full training with only 4k-shot transfer.

## 6.2 RESULTS AND DISCUSSION

**Comparison between our solutions and existing alternatives (Q1).** Table 2 reports execution times across neural network architectures on four GPUs. DOPPLER-SYS adopts all three training stages described in Section 5, and DOPPLER-SIM adopts only Stages I and II. The results show that DOPPLER-SYS outperforms all baselines in most settings, with DOPPLER-SIM often ranking second best. For example, DOPPLER-SYS reduces runtime by up to 78.2% over CRITICAL PATH,

62.5% over PLACETO, and 52.7% over GDP, while both DOPPLER-SYS and DOPPLER-SIM surpass ENUMERATIVEOPTIMIZER by up to 13.8% (see ablation studies on random seeds in Appendix G.2).

**Ablation study for select and place policies (Q2).** Table 3 presents an ablation study isolating the effects of the node and device policies. In the ablated variants, we replace our policies with CRITICAL PATH strategies: DOPPLER-SEL assigns selected nodes to the earliest-available device, while DOPPLER-PLC selects nodes with the longest path to an exit. Overall, combining both policies yields the best performance. For CHAINMM, DOPPLER-PLC slightly outperforms DOPPLER-SYS by a few milliseconds, but for more complex models, the combined policies provide clear gains.

**Improving training using Stage I, II, and III (Q3).** Fig. 4 shows performance for DOPPLER-SYS when trained using different combinations of imitation learning (I), simulations (II), and real-system executions (III) for the LLAMA-LAYER dataflow graph. As we hypothesized, training on the real-system only leads to slower convergence due to the need to explore from a poor initial model, resulting in unstable performance. Imitation learning and simulations enable faster convergence and lower execution times. See ablation studies on pretraining PLACETO in Appendix G.4.

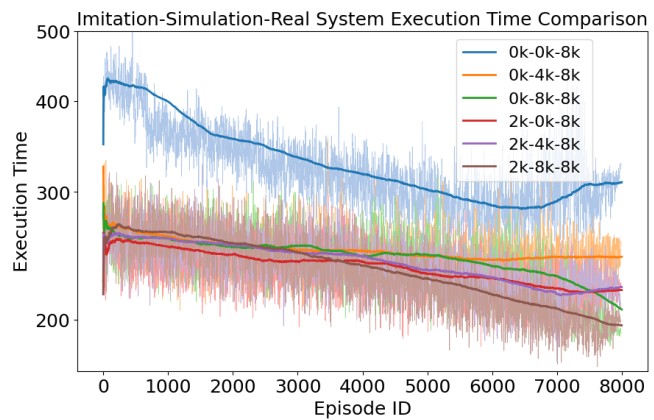

Figure 4: Real engine execution times (in milliseconds) for DOPPLER-SYS using different combinations of three training stages for the LLAMMA-LAYER dataflow graph.

**Visualizing DOPPLER-SYS's assignments (Q4).** Fig. 5 shows the assignments produced by DOPPLER-SYS for the FFNN dataflow graph, which achieves both GPU load balancing and communication minimization along the critical path. By further profiling how the assignments are scheduled in the system (see Appendix C), we find that DOPPLER's assignments often enable overlapping communication and computation across GPUs, minimizing stalling and maximizing utilization. We provide additional studies in Appendix A, including detailed assignment visualizations, GPU profiling results, and analyses of the node-selection order generated by SEL.

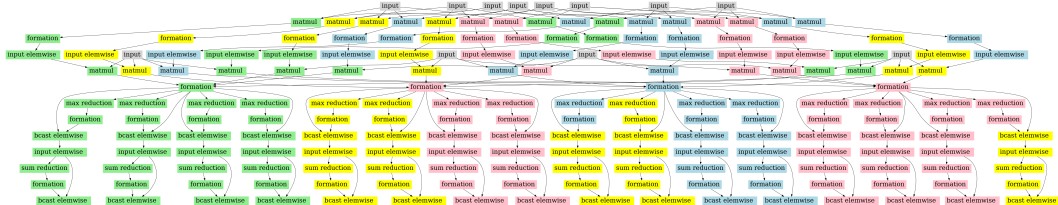

Figure 5: Assignments for FFNN produced by DOPPLER. Colors indicate the mapping of computations to GPUs. DOPPLER effectively balances load while minimizing inter-GPU communication.

**DOPPLER's transfer ability across graphs and architectures (Q5).** We evaluate transfer learning (1) from simple graph (FFNN, CHAINMM) to Llama-structured graphs on the same hardware in Table 4 and (2) across hardware architectures for the same graph. With 2K fine-tuning episodes, DOPPLER adapts to new architectures and outperforms baselines. With 4K episodes (less than half of the original training), it finds assignments comparable to full target training. In our hardware adaptability experiments, we train a policy for the FFNN graph with four P100 GPUs and transfer it to eight V100 GPUs. The zero-shot setting yields 82.7% of communication intra-GPU, 6.7% within the same GPU group (with all-to-all NVLink), and 10.6% across GPUs without direct NVLink. After 2K episodes, the policy improves assignments to 94.7% ($\uparrow 12.0\%$) intra-GPU, 1.9% ($\downarrow 4.8\%$) within NVLink groups, and only 3.4% ($\downarrow 7.2\%$) across GPUs without NVLink (detailed results in Appendix J), which justifies performance improvement.

**DOPPLER training and inference scalability (Q6).** We analyze DOPPLER's scalability in both training and inference time for dataflow graphs with increasing size in Fig. 6. The figure shows that

DOPPLER scales linearly with the size of graphs and achieves the lowest training and inference times, compared to RL-based baselines such as GDP. In terms of performance for much larger dataflow graphs, they are not substantially more complex than our experiment setups, as modern large-scale training and inference rely heavily on data parallelism (Li et al., 2020b) and pipelining (Narayanan et al., 2021). More discussions are provided in Appendix I.

## 7  RELATED WORKS

**Classical Approaches for Device Placement and Scheduling.** List scheduling (LS) heuristics, such as CRITICAL PATH, decompose the problem of computing a schedule into a sequence of *select* and *place* steps (Kwok & Ahmad, 1999). DOPPLER can be seen as a neural LS heuristic that learns to *select* and *place* directly from observations using an MDP. Our experiments show that DOPPLER outperforms CRITICAL PATH. Graph partitioning (Kernighan & Lin, 1970; Kirkpatrick et al., 1983; Fiduccia & Mattheyses, 1988; Johnson et al., 1989; Hagen & Kahng, 1992; Karypis, 1997) can also be applied to device placement, but previous work has shown that RL is a better alternative for this problem (Mirhoseini et al., 2017).

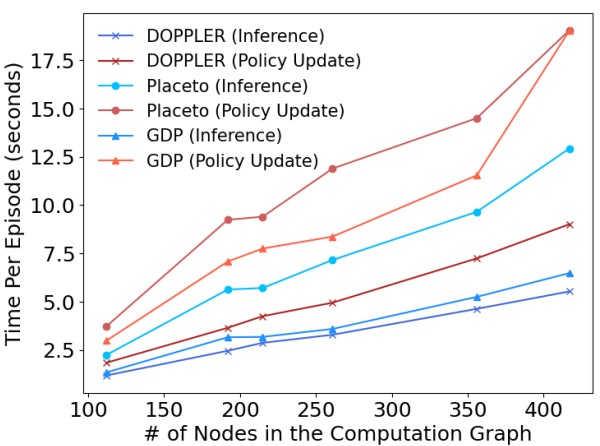

Figure 6: Inference time and RL policy update time as the number of nodes in the dataflow graph increases.

**Reinforcement Learning for Combinatorial Optimization.** Traditional algorithms for combinatorial optimization problems often rely on hand-crafted heuristics that involve sequentially constructing a solution. Recently, there has been growing interest in applying RL (and deep learning more broadly) to learn heuristics for these problems. RL can learn constructive sequential heuristics for large combinatorial problems and exploit structural similarity between related instances, enabling generalization (Mazyavkina et al., 2021). For instance, (Bello et al., 2017) introduced a policy gradient method for the Traveling Salesman Problem (TSP). Subsequent studies extended RL to problems beyond TSP (Khalil et al., 2017; Cappart et al., 2019; Drori et al., 2020; Emami & Ranka, 2018; Lu et al., 2019; Mazyavkina et al., 2021; Nazari et al., 2018; Kool et al., 2019; Abe et al., 2019; Manchanda et al., 2020; Chen & Tian, 2019; Li et al., 2020a; Laterre et al., 2018; Gu & Yang, 2020; Cai et al., 2019). Our work is unique in leveraging combinatorial structure, list-scheduling heuristics, and direct access to the target system during training to address the device assignment problem using RL.

**RL for Device Placement and Scheduling.** Early work introduced a sequence-to-sequence RNN trained with policy gradients for device placement, showing that RL can outperform heuristics (Mirhoseini et al., 2017; Pellegrini, 2007). PLACETO (Addanki et al., 2019) replaced the RNN with a GNN, while Paliwal et al. (2019) combined RL with a genetic algorithm, though at high evaluation cost. Zhou et al. (2019) proposed a graph-embedding approach with sequential attention and a single placement policy. DOPPLER introduces dual-policy learning with three-stage training for faster convergence and continuous optimization. More recent work includes end-to-end optimization from graph construction to placement (Duan et al., 2024) and improved node representations using cosine phase position embeddings (Han et al., 2024), which are complementary to our approach.

## 8  CONCLUSION

In this paper, we have considered the problem of assigning computations in a dataflow graph to devices to minimize execution time in a work-conserving system. We have proposed DOPPLER, a dual-policy learning framework for learning device assignment in three stages. Some of the key innovations are (1) DOPPLER explicitly tries to learn an approximate node traversing order, to make the assignment problem easier, (2) DOPPLER adopts two pre-training stages using imitation learning and simulation-based learning to speed up policy convergence, and (3) DOPPLER continues dual-policy training during deployment, achieving a gradual reduction in execution time over time.

## REPRODUCIBILITY STATEMENT

We provide details of experimental settings and hyperparameters used for training in Section 6.1 to reproduce our results and experiments. We have included our code and data used for running all experiments at https://github.com/xinyuyao/Doppler.

## ACKNOWLEDGMENT

We acknowledge the support by the US Department of Transportation Tier-1 University Transportation Center (UTC) Transportation Cybersecurity Center for Advanced Research and Education (CYBER-CARE) (Grant No. 69A3552348332), by NSF grants 1918651, 2008240, 2131294, and 2212557, by the NIH under CTSA #UM1TR004906 and by the Rice Ken Kennedy Institute.

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

We provide a brief roadmap of the appendix below:

**Appendix A**: Device assignment analysis with system profiling and visualizations in graphs across methods.

**Appendix B**: Enumerative Assignment Algorithm (Algorithm 4) for our baseline ENUMER-ATIVE OPTIMIZER

**Appendix C**: Implementation of the system we used in Stage III training and for evaluating execution time for assignments reported in tables.

**Appendix D**: Detailed configurations for computation graphs in experimental setups.

**Appendix E**: Detailed Graph Neural Network features used in dual-policy networks.

**Appendix F**: Configurations for synchronous system runs in Table 1.

**Appendix G**: More ablation studies on 1) simulator (Stage II training and DOPPLER-SIM), 2) reinforcement learning trainings across random seeds, 3) efficiency performance trade-off for message passing on Doppler, 4) pretrain placeto with Stage I.

**Appendix H**: Experiments on various hardware configurations, including various GPU memory sizes and different numbers and types of GPUs.

**Appendix I**: Discussion on scaling to much larger dataflow graphs and larger numbers of GPUs.

**Appendix J**: Doppler's transfer ability across hardware with analysis.

## A  MORE DEVICE ASSIGNMENT ANALYSIS AND VISUALIZATIONS

### A.1  COMPUTATION NODE DETAILS

- **input:** input tensors
- **matmul:** matrix multiplications on two matrices
- **input elemwise:** elementwise operations (e.g., ReLU) on an input tensor
- **straight elemwise:** elementwise operations (e.g., ReLU) with two inputs having the same dimensions
- **bcast elemwise:** takes two inputs of different shapes (e.g., a matrix and a vector), performing an elementwise operation (e.g., ReLU) with one element of the vector applied to an entire row of the matrix
- **max reduction:** reduce one dimension by finding the maximum.
- **min reduction:** reduce one dimension by finding the minimum.
- **sum reduction:** reduce one dimension by finding the sum along that dimension.
- **product reduction:** reduce one dimension by finding the product along that dimension.
- **formation:** a placeholder operation that forces aggregations in meta-ops groups (defined in Appendix B) to form a single tensor.
- **complexer:** a conversion between floating-point and complex tensors.
- **fill:** an operation to create tensors with all elements set to the same scalar, or to assign all lower or upper diagonal elements with provided scalar values.
- **squeezer:** adding or removing singleton dimensions of a tensor.
- **selec:** an operation to copy a subset of several input tensors into an output tensor—a generalization of tensor subset and tensor concatenation.

### A.2  ASSIGNMENT PROFILING

This section examines the performance results from Table 2, where the DOPPLER algorithm achieved a lower runtime compared to CRITICAL PATH, PLACETO, GDP, and the expert-designed ENUMERATIVEOPTIMIZER.

During the development of the ENUMERATIVEOPTIMIZER algorithm, it became clear that for meta-ops (described in Appendix B) containing many computations, the devices should be fully utilized and load-balanced. **Put more succinctly, it is expected that good assignments should minimize data transfers while maximizing computational resource utilization.** In the following, we provide detailed analysis into CHAINMM and FFNN throught assignment visualizations and system profiling.

### A.2.1  CHAINMM

The assignments in Fig. 7 for DOPPLER show all four devices (four colors of nodes) being used, whereas for ENUMERATIVEOPTIMIZER, in Fig. 8, only two of the devices are used for the latter computations. The corresponding device utilization plots are shown in Fig. 9 and Fig. 10, respectively. It appears that, indeed, ENUMERATIVEOPTIMIZER does not fully utilize available compute resources toward the end of the computation (after 80ms). In contrast, DOPPLER performs well from the beginning to the end, as shown in Fig. 7.

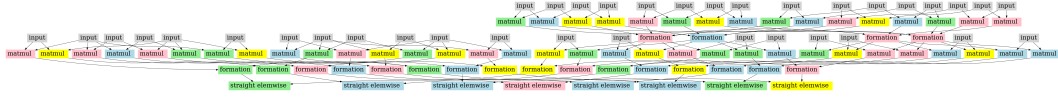

Figure 7: Assignment found by DOPPLER for ChainMM

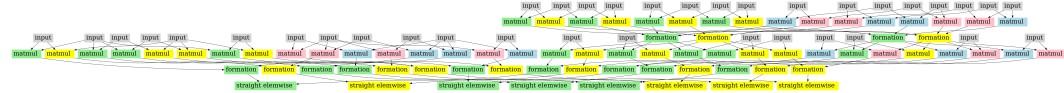

Figure 8: Assignment found by ENUMERATIVEOPTIMIZER for ChainMM

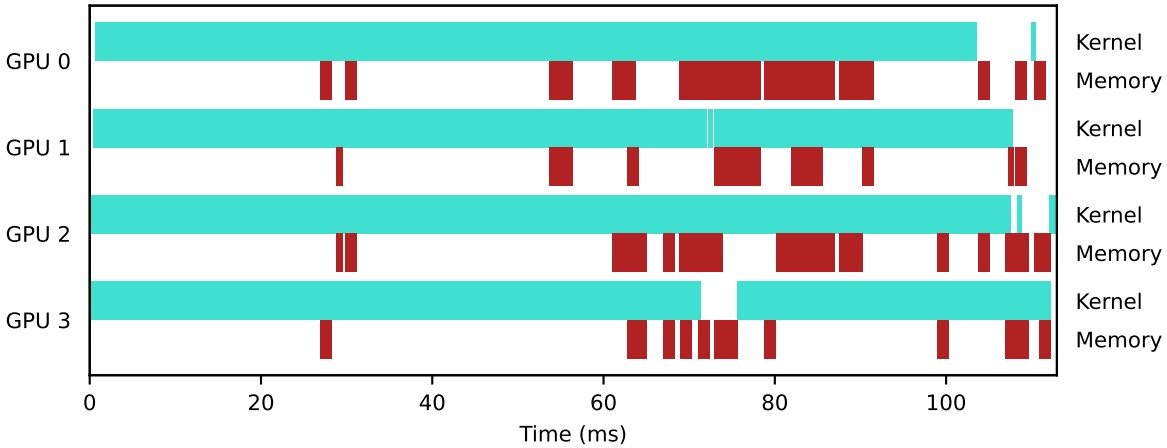

Figure 9: Device and transfer utilization for DOPPLER, CHAINMM.

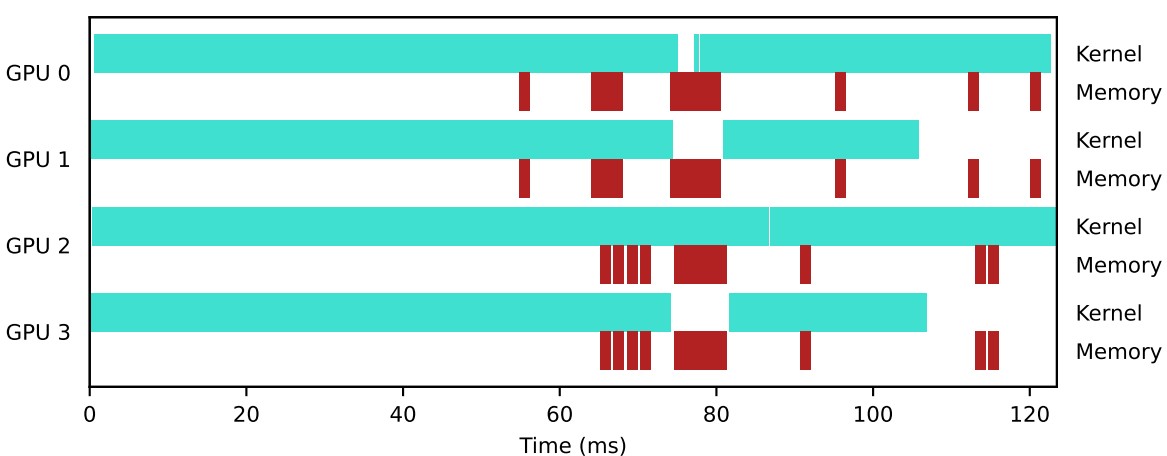

Figure 10: Device and transfer utilization for ENUMERATIVEOPTIMIZER, CHAINMM.

### A.2.2 FFNN

The assignments for FFNN with DOPPLER and PLACETO are shown in Fig. 11 and Fig. 12; the corresponding device utilization plots are shown in Fig. 13 and Fig. 14. In the DOPPLER assignments, subsequent vertices typically share the same device assignment. In turn, this leads to a lower amount of data transfer. For the PLACETO assignments, however, subsequent vertices do not tend to have the same device assignment, possibly leading to a large amount of data transfer. In the device utilization figures, this is indeed borne out: the PLACETO execution is three times slower, with most of the time spent on data transfers across GPUs.

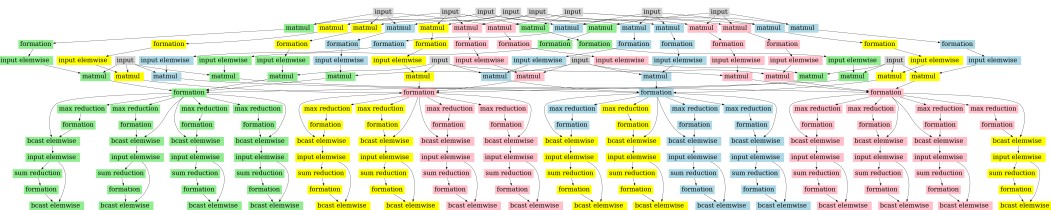

Figure 11: Assignments found by DOPPLER for FFNN.

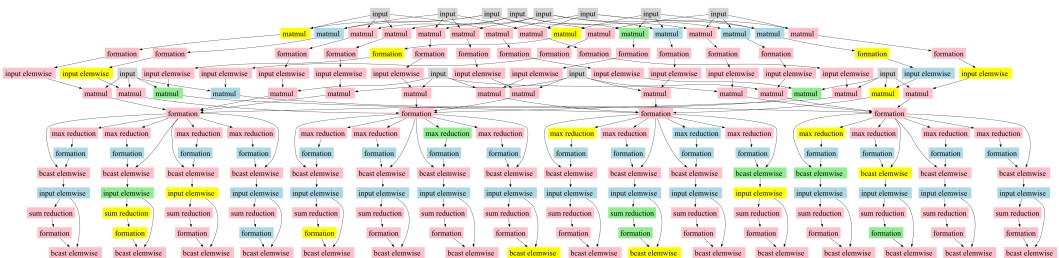

Figure 12: Assignments found by PLACETO for FFNN.

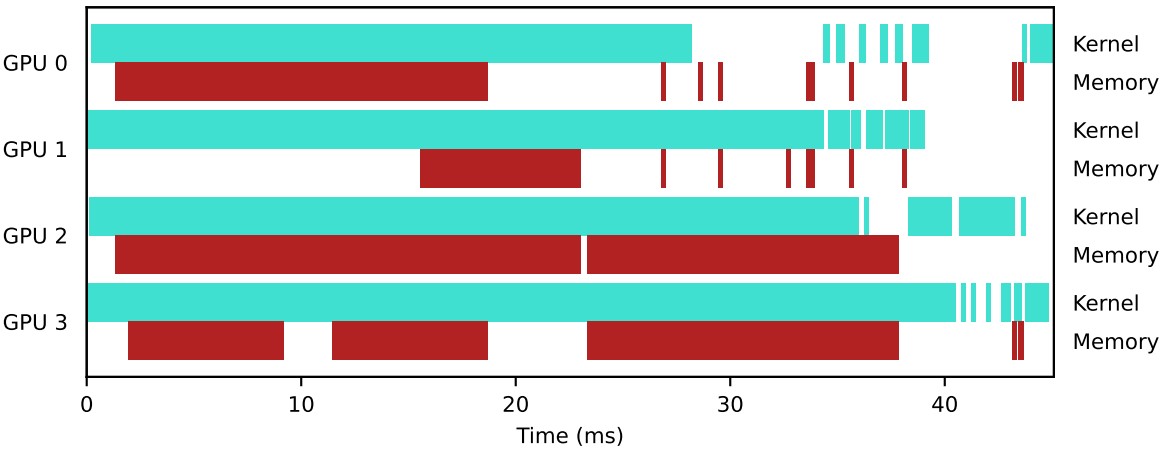

Figure 13: Device and transfer utilization for DOPPLER, FFNN.

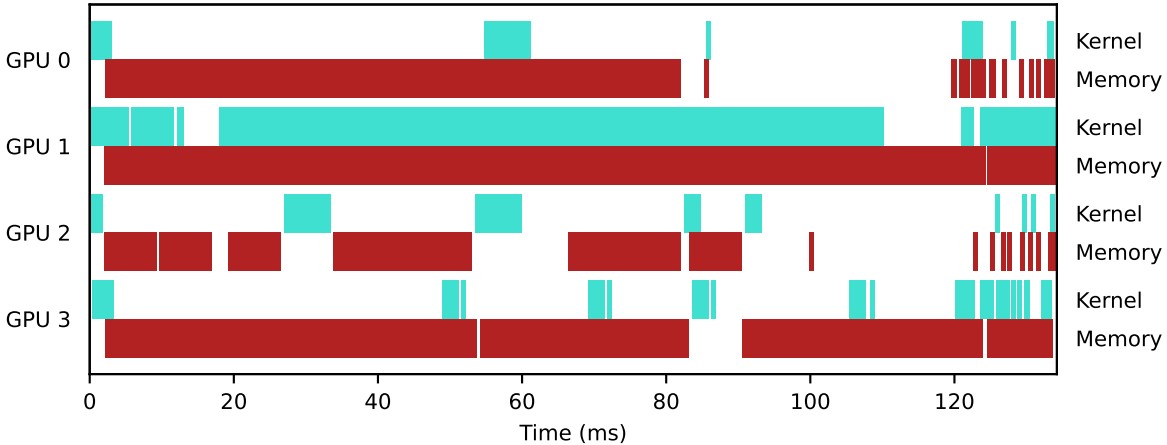

Figure 14: Device and transfer utilization for PLACETO, FFNN.

### A.3 LLAMA-BLOCK ASSIGNMENT ANALYSIS WITH RESPECT TO NODE-SELECTION ORDER

We showed the assignments with respect to the selected node order in Fig. 15, Fig. 16, Fig. 17, Fig. 18, Fig. 19. We observed two interesting patterns that $\text{SEL}_\theta$ follows.

The first pattern is that $\text{SEL}_\theta$ performs level-wise node assignment to maintain load balancing. In Fig. 15, Fig. 16, and Fig. 17, each matrix multiplication, tensor formation, and element-wise input operation is selected sequentially and distributed evenly across the four GPUs. This strategy keeps all GPUs fully utilized during this phase of execution. **At this stage, the workload is computation-bound, meaning many operations are ready to execute in parallel. As a result, maximizing GPU utilization by distributing computations evenly is more important than minimizing communication overhead**—a trade-off that competing baselines fail to manage effectively.

The second pattern is that $\text{SEL}_\theta$ prioritizes selecting nodes for which it has high confidence in determining an optimal device placement. In Fig. 18 and Fig. 19, **the system becomes network-bound, with limited parallel computations available and communication bandwidth emerging as the primary bottleneck.** In this regime, it is more beneficial to select nodes along the same execution path and assign them to the same GPU to reduce cross-device communication, rather than distributing independent nodes across different GPUs.

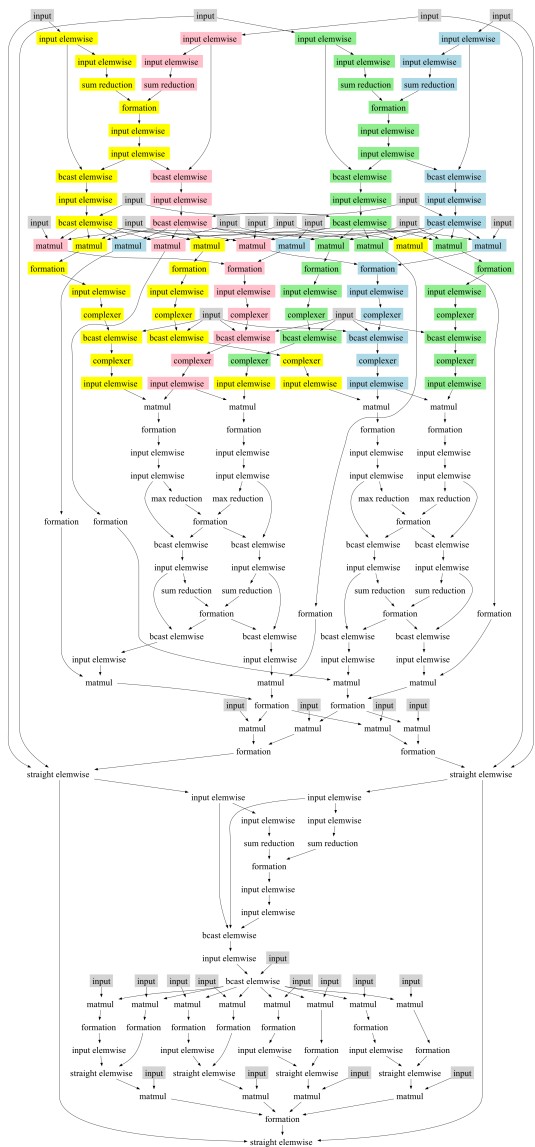

Figure 15: Assignment found by DOPPLER for LLAMA-BLOCK at **Step 78**

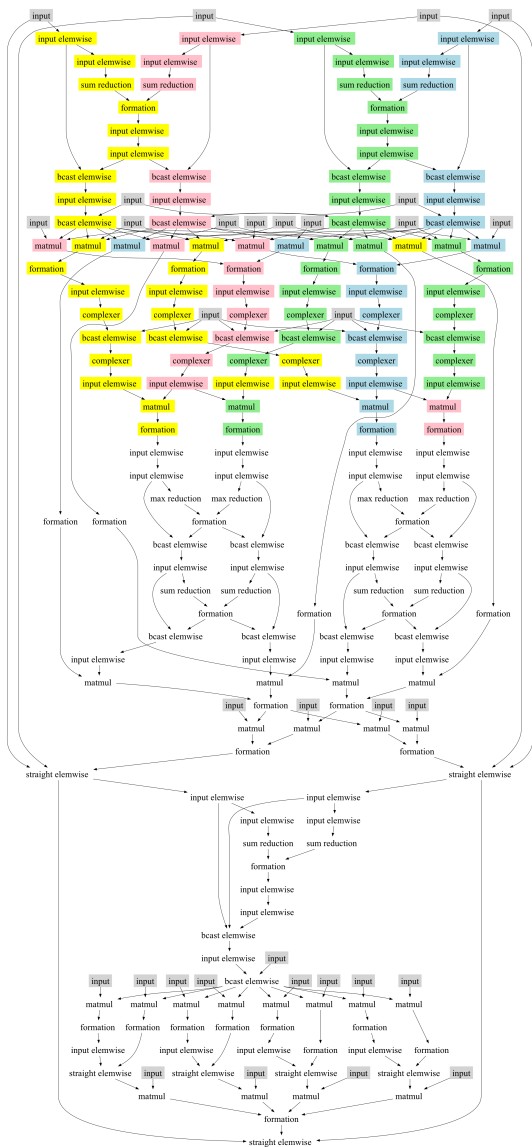

Figure 16: Assignment found by DOPPLER for LLAMA-BLOCK at **Step 86**

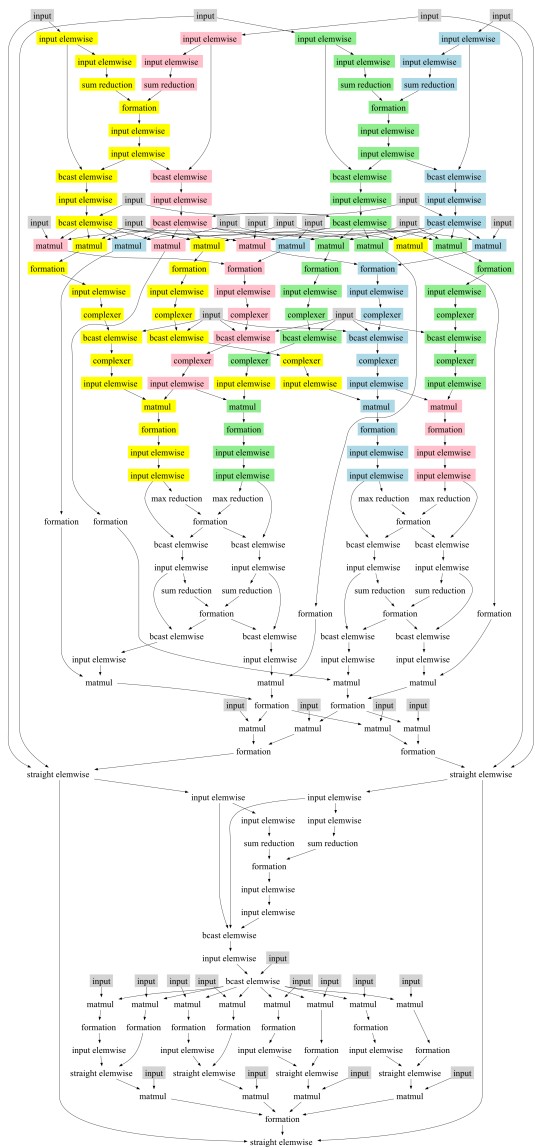

Figure 17: Assignment found by DOPPLER for LLAMA-BLOCK at **Step 94**

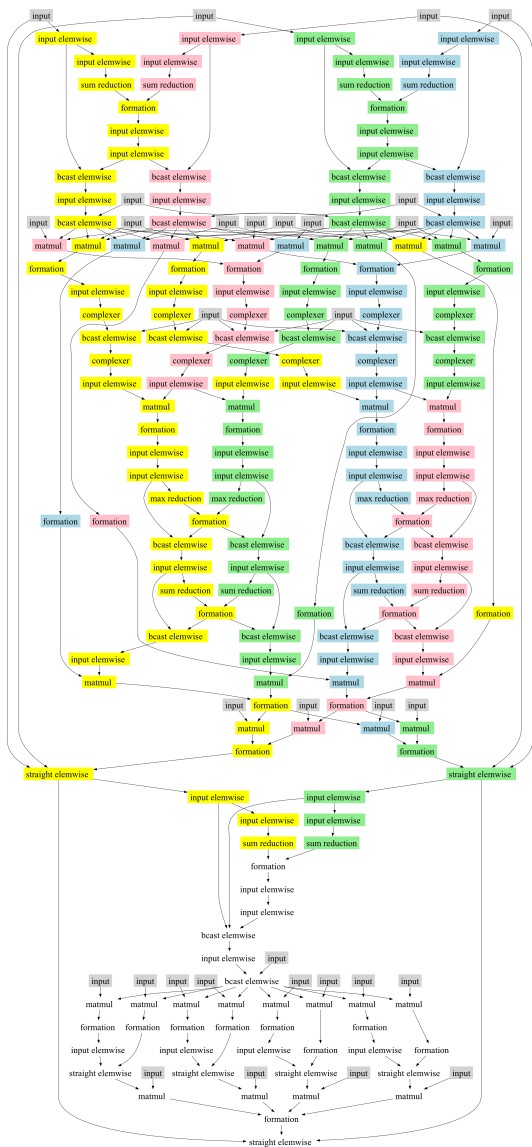

Figure 18: Assignment found by DOPPLER for LLAMA-BLOCK at **Step 146**

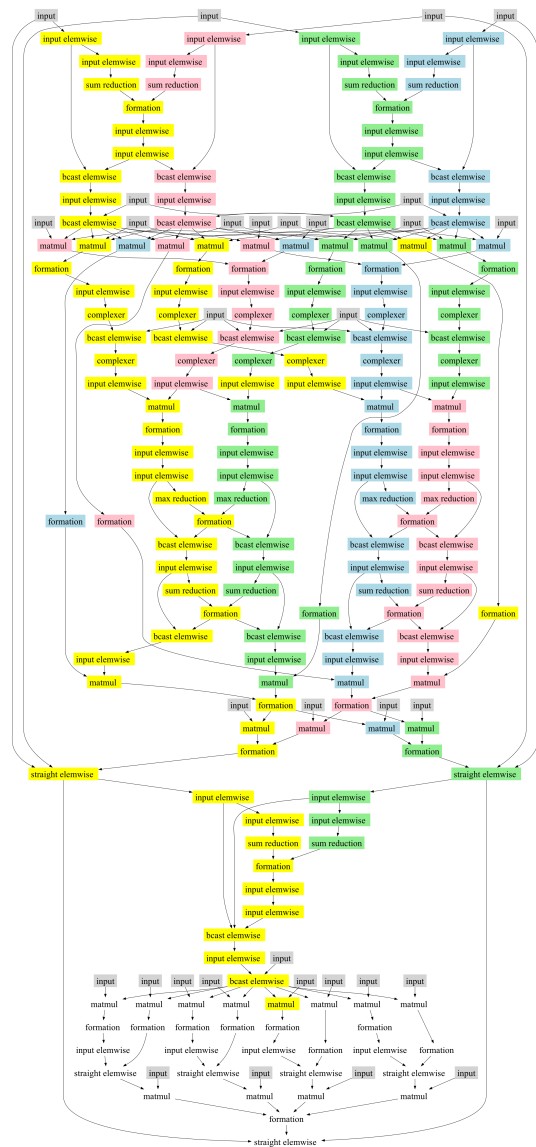

Figure 19: Assignment found by DOPPLER for LLAMA-BLOCK at **Step 153**

## A.4 LLAMA-LAYER ASSIGNMENT VISUALIZATIONS

We show the best assignment we found for five methods on the LLAMA-LAYER computation graph: DOPPLER (Fig. 20), ENUMERATIVEOPTIMIZER (Fig. 21), PLACETO (Fig. 22), CRITICAL PATH (Fig. 23), and GDP (Fig. 24).

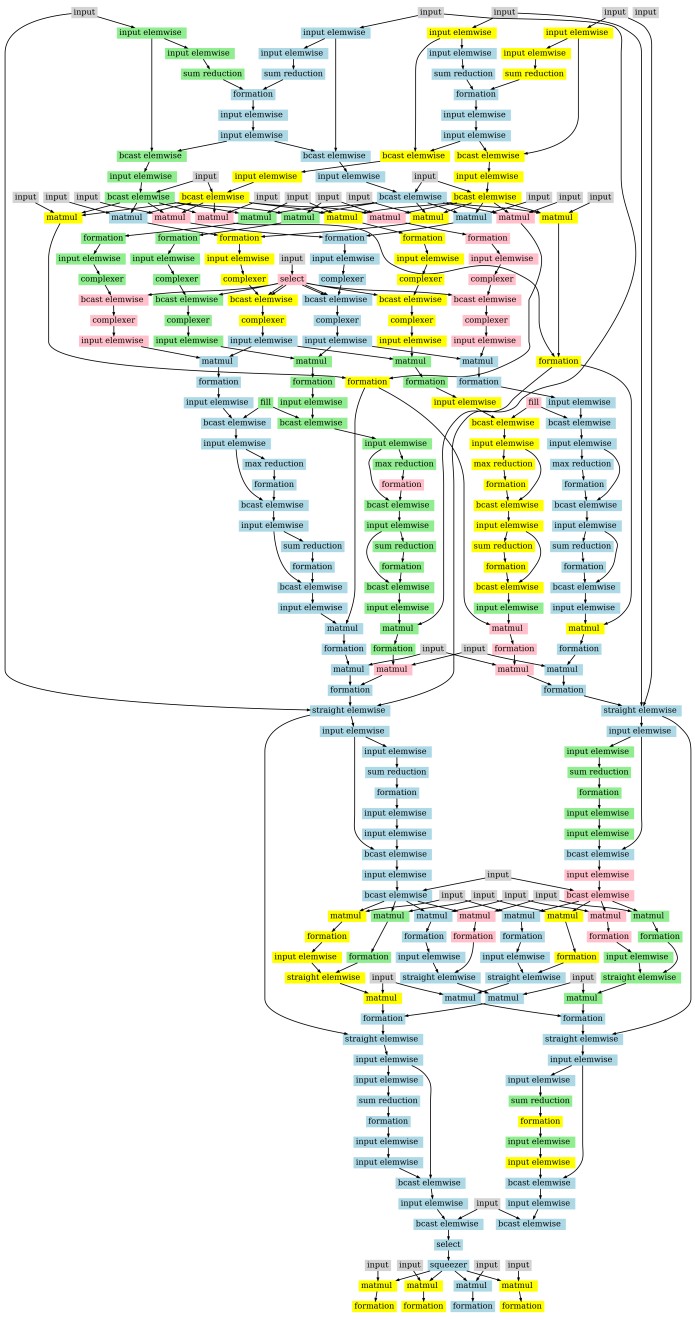

Figure 20: Assignment found by DOPPLER for LLAMA-LAYER

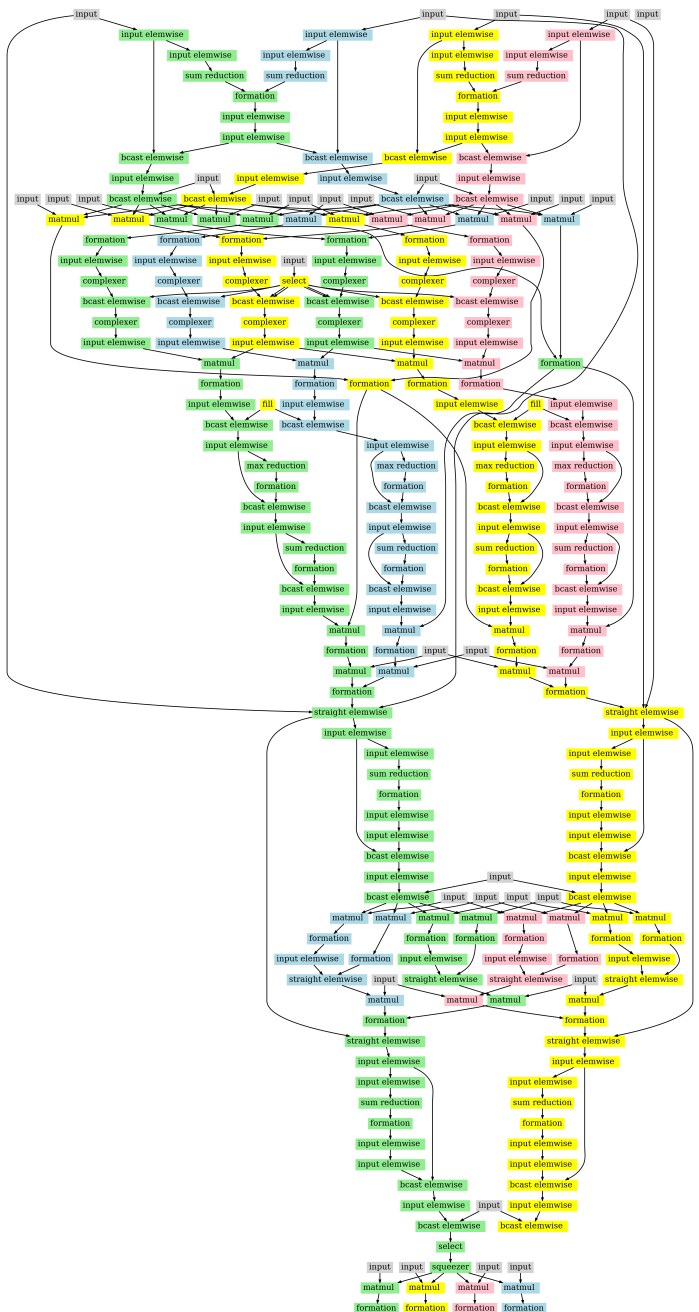

Figure 21: Assignment found by ENUMERATIVEOPTIMIZER for LLAMA-LAYER

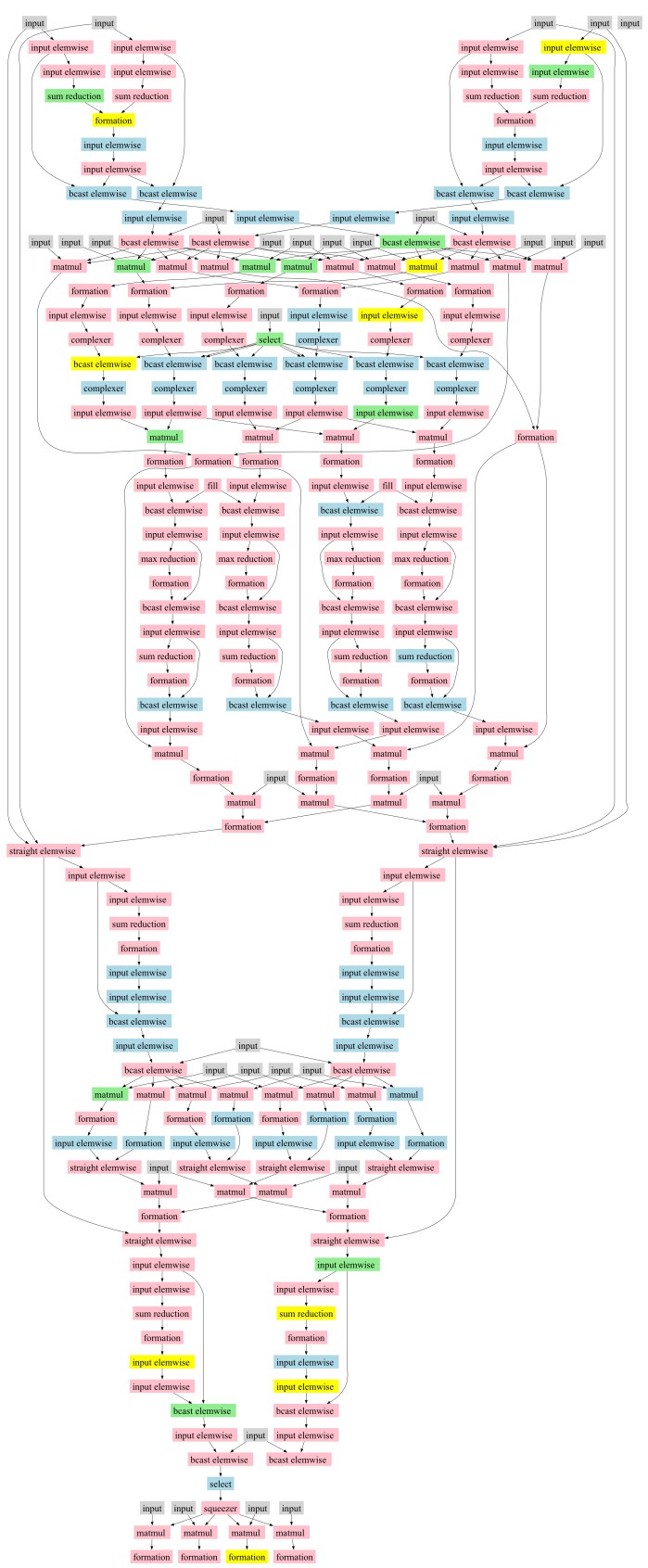

Figure 22: Assignment found by PLACETO for LLAMA-LAYER

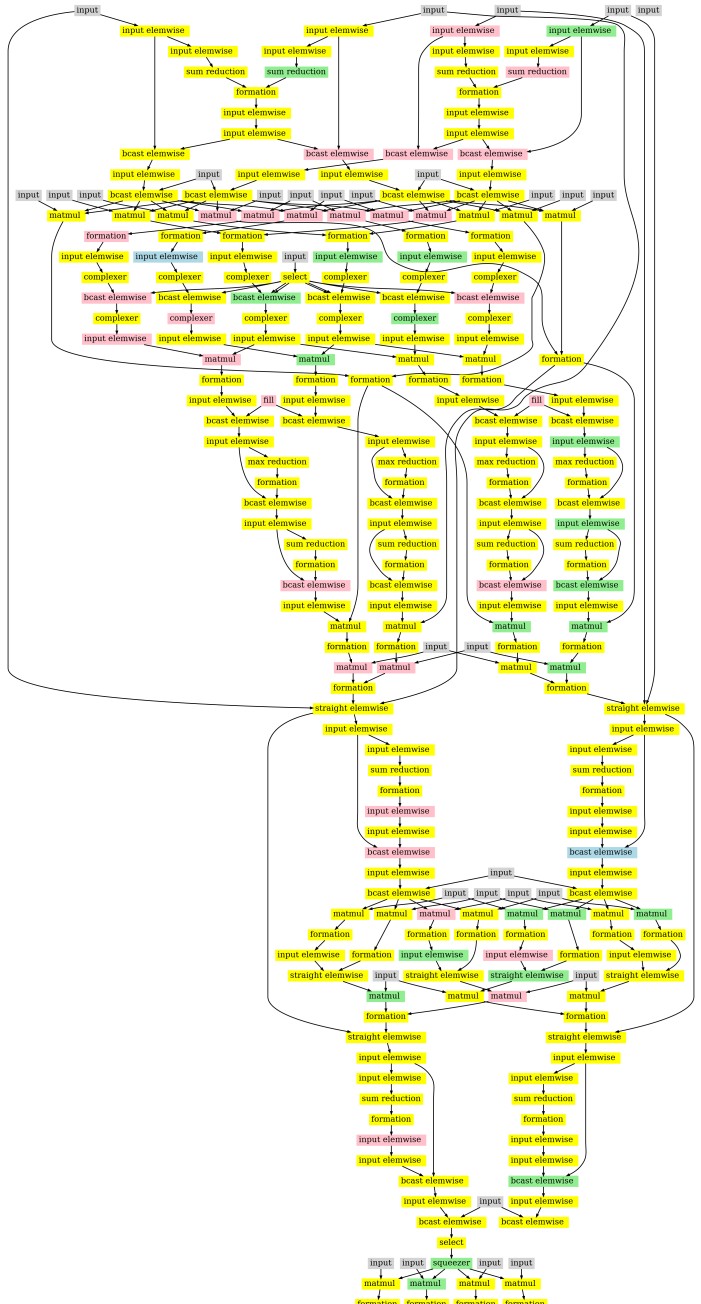

Figure 23: Assignment found by CRITICAL PATH for LLAMA-LAYER

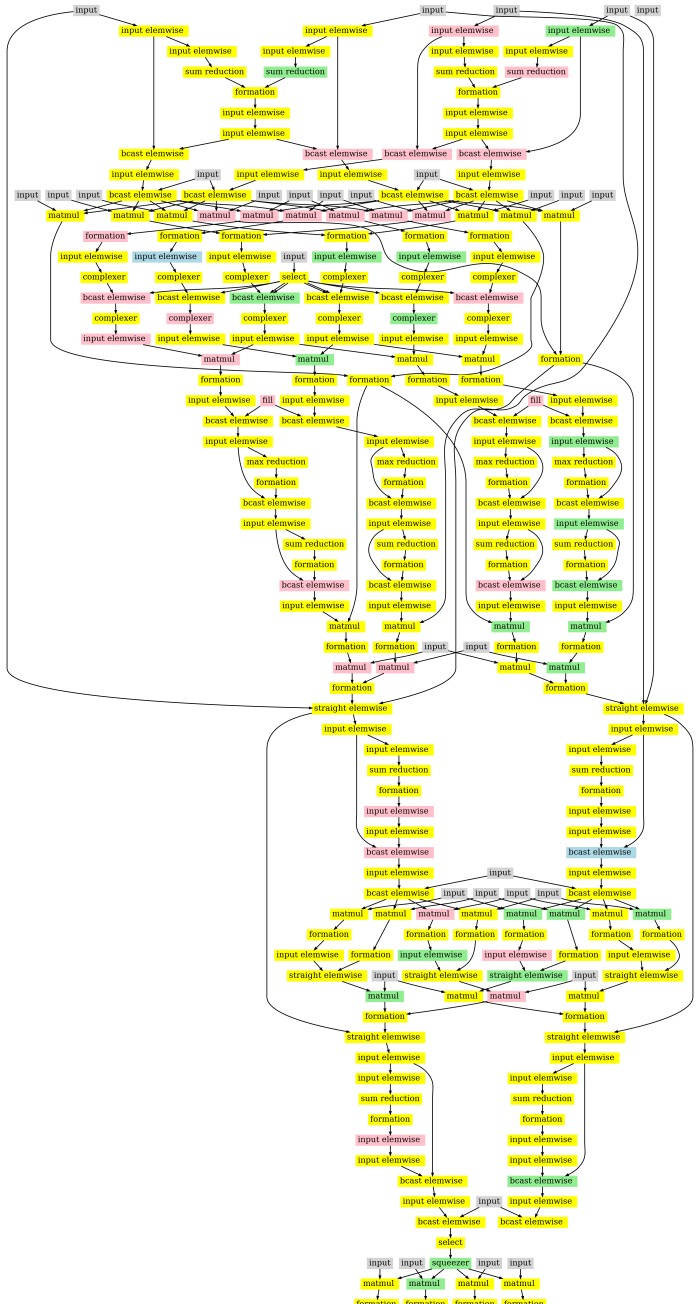

Figure 24: Assignment found by GDP for LLAMA-LAYER

# B  ENUMERATIVE ASSIGNMENT ALGORITHM (ALGORITHM 4)

**TL;DR. The Enumerative Assignment Algorithm is a greedy, sharded-ops–by–sharded-ops device placement strategy that traverses the graph level-by-level, exhaustively trying shard assignments to minimize estimated communication cost, without global lookahead or learning.**

In this section, we describe our enumerative assignment algorithm, which uses a level-by-level, exhaustive enumeration in an attempt to find an assignment $A$ for the vertices in a graph $\mathcal{G}$ that minimizes $\texttt{ExecTime}(A)$. This algorithm uses a greedy approach that first groups vertices based on the graph structure and then attempts a subset of possible assignments for the operations within each group, selecting the assignment with the minimum estimated cost.

This algorithm requires that the vertices in the graph $\mathcal{G}$ have been organized into a list of "meta-ops" called $M$. As our input dataflow graph has been created by sharding a compute graph using a framework such as Alpa Zheng et al. (2022), each operation in the input dataflow graph $\mathcal{G}$ is descended from some operation that has been sharded. For example, consider Fig. 1. All eight `MMul` ops at the lowest level of the graph, as well as the four `MAdd` ops at the next level, were created by sharding $X \times Y$. We group all twelve of these operations and term them a "meta-op." Further, we can topologically order these meta-ops so that if $m_1$ comes before $m_2$ in $M$, it means that none of the vertices in $m_2$ can be reached from any vertex in $m_1$ by traversing $\mathcal{E}$.

Note that each meta-op has two subsets—one of which may be empty: a set of computationally expensive operations (such as the `MMul` ops) that result directly from sharding the original operation, and a set of less expensive operations needed to aggregate and/or recompose the results of the first set of operations. For a meta-op $m$, we call these $m.\texttt{shardOps}$ and $m.\texttt{reduceOps}$. Note that the original operation is always sharded so that if there are $n$ devices, there are $n$ items in $m.\texttt{shardOps}$, and load balancing of these shards is crucial. Thus, our tactic is to always partition $m.\texttt{shardOps}$ across the $n$ devices and never assign two operations in $m.\texttt{shardOps}$ to the same device. Likewise, if the meta-op is sharded into $n$ parts, there will always be at most $n$ items in $m.\texttt{reduceOps}$. Therefore, we always partition them across (possibly a subset of) the devices.

Given this, our algorithm, ENUMERATIVEOPTIMIZER, proceeds through the list $M$ of meta-ops in order. For each $m$, it exhaustively tries all assignments of $m.\texttt{shardOps}$. Each assignment is costed by computing the time required to transfer all of the items in $m.\texttt{shardOps}$ to where they will be consumed. These times are estimated using statistics gathered by testing transfers on the actual hardware. Once $m.\texttt{shardOps}$ is placed, $m.\texttt{reduceOps}$ is placed using the same cost model. Because of the ordering of the meta-ops in $M$, and because we process $m.\texttt{shardOps}$ before $m.\texttt{reduceOps}$, we always know the assignment of the inputs to $m.\texttt{shardOps}$ or $m.\texttt{reduceOps}$ before placing them, and thus it is easy to compute the cost.

This algorithm is greedy in the sense that it processes meta-ops one at a time, from start to finish, using the topological ordering. Thus, if the cost model is correct, it will be optimal only as long as each $m.\texttt{shardOps}$ and $m.\texttt{reduceOps}$ is run in lock-step, with a barrier before each set is executed. Obviously, this is not the case in reality with a dynamic scheduler, but one might expect the algorithm to produce a good assignment in practice.

---

**Algorithm 4** EnumerativeOptimizer

---

**Input:** Sorted list of meta-ops $M$ containing all vertices in $\mathcal{G}$
**Output:** Assignment $A$
A $\leftarrow \langle \rangle$ % assignment is empty
% loop thru meta-ops
**for** $m \in M$ **do**
  % First deal with the shared op
  $A \leftarrow$ getBestAssign($m$.shardOps, $A$)
  % now place the reductions
  $A \leftarrow$ getBestAssign($m$.reduceOps, $A$)
**end for**
return $A$

% computes the best assignment for a set of vertices that are expected to run in parallel on all devices
**subroutine** getBestAssign($vertices, A$)
$bestCost \leftarrow \infty; bestAssign \leftarrow$ null
% loop through all possible device assignments
**for** $D \in$ allPerms($\mathcal{D}$) **do**
  $whichDev \leftarrow 0; cost \leftarrow 0$
  % loop through the ops created by sharding this meta-op
  **for** $v \in vertices$ **do**
    % loop through inputs to this op and add network cost
    **for** $v_1 \in \mathcal{V}$ s.t. $(v_1, v) \in \mathcal{E}$ **do**
      $cost \leftarrow cost +$ getNetworkTime($v_1, a_{v_1}, D_{whichDev}$)
    **end for**
    $whichDev \leftarrow whichDev + 1$
  **end for**
  **if** $cost < bestCost$ **then**
    $bestCost \leftarrow cost$
    $bestAssign \leftarrow D$
  **end if**
**end for**
% we have the best assignment for this meta op, so record it
$whichDev \leftarrow 0$
**for** $v \in vertices$ **do**
  $a_v \leftarrow D_{whichDev}$
  append $a_v$ to $A$
  $whichDev \leftarrow whichDev + 1$
**end for**
return $A$

---

## C  SYSTEM IMPLEMENTATION (DOPPLER-EINSUMMABLE)

The underlying system runtime Bourgeois et al. (2025); Ding et al. (2024) that executes our graphs is written in C++. The dataflow graph $\mathcal{G}$ is executed asynchronously by a single-threaded event loop whose job is to monitor when the dependencies implicit in the graph are satisfied. When the inputs to a vertex are found to be available, the event loop checks whether the resources necessary to execute the vertex are also available (a "resource" may be an open GPU stream or an open communication channel). If resources are available, the event loop may choose to execute the vertex. We use the term "event loop" because the loop requests the necessary resources to execute the vertices in $\mathcal{G}$ in response to "events." An event occurs whenever a graph dependency is satisfied or when a previously used resource becomes available. The main event loop is notified of this via an asynchronous callback. In our implementation, we use CUDA version 11.8.0 and cuTensor version 2.0.1.

## D  COMPUTATION GRAPHS USED IN THE EXPERIMENTS

This section describes the computation graphs used in our experiments, which are designed to capture a range of workload characteristics encountered in modern ML systems. We include (i) ChainMM, a matrix-multiplication graph with long dependency chains and parallel substructures; (ii) FFNN, a feed-forward neural network modeling dense linear layers and nonlinear activations typical of deep learning workloads; and (iii) Llama-block / Llama-layer, which represent transformer-style computation graphs from large language models. Together, these graphs span diverse operator mixes, graph sizes, and dependency patterns, enabling a comprehensive evaluation of DOPPLER across synthetic, neural, and large-model workloads.

### D.1  CHAINMM

- **Inputs**: Five matrices $A, B, C, D, E \in \mathbb{R}^{10000 \times 10000}$.
- **Computation**: $(A \times B) + (C \times (D \times E))$.
- **Graph size**: 112 nodes.

### D.2  FFNN

- **Inputs**:
  - Input batch matrix: $X \in \mathbb{R}^{2^{15} \times 2^5}$.
  - First-layer weights: $W^{(1)} \in \mathbb{R}^{2^5 \times 2^{16}}$.
  - First-layer bias: $b^{(1)} \in \mathbb{R}^{2^{16}}$.
  - Output-layer weights: $W^{(2)} \in \mathbb{R}^{2^{16} \times 2^5}$.
  - Output-layer bias: $b^{(2)} \in \mathbb{R}^{2^5}$.
- **Computation**:
  - Hidden layer:
  $$H = \text{ReLU}\left(XW^{(1)} + b^{(1)}\right), \quad H \in \mathbb{R}^{2^{15} \times 2^{16}}.$$
  - Output layer:
  $$Y = \text{Softmax}\left(HW^{(2)} + b^{(2)}\right), \quad Y \in \mathbb{R}^{2^{15} \times 2^5}.$$
- **Activations**: ReLU for the hidden layer; Softmax for the output layer.
- **Graph size**: 192 nodes.

### D.3  LLAMA-BLOCK AND LLAMA-LAYER

- **Model configuration**:
  - Number of parameters: 7B.
  - Embedding dimension: 4096.

- Maximum sequence length: 4096.
- Vocabulary size: 32,000 tokens.
- Batch size: 1.
- Number of layers: 1.

- **Graph size**: 215 nodes.

- **Computation**: The computation graph follows the standard Llama transformer block and layer structure, as illustrated in Fig. 25.

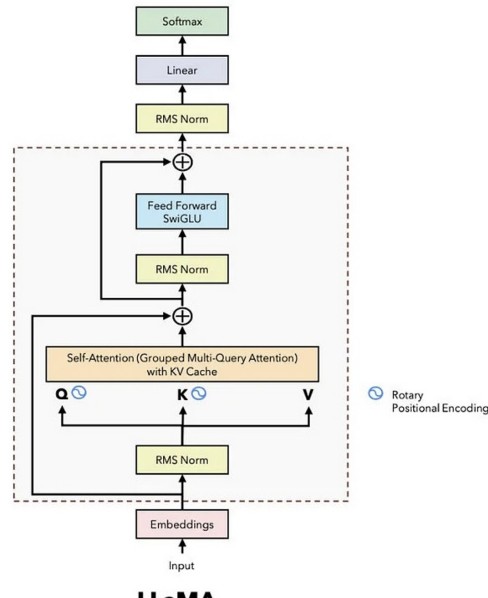

Figure 25: Llama-block and Llama-layer architecture. Figure from Yaadav (2024)

# E  GRAPH FEATURES $X_{\mathcal{G}}$ AND DEVICE FEATURES $X_{\mathcal{D}}$

Given a computation graph $\mathcal{G} = (\mathcal{V}, \mathcal{E})$ with $\mathcal{V} = \{v_1, v_2, \ldots, v_n\}$ and $\mathcal{E} = \{e_1, e_2, \ldots, e_m\}$, we define the following:

- **Computation cost for $v$.** We use the floating-point operations per second of $v$ as the computation cost.
- **Communication cost for $e_{i,j} = (v_i, v_j)$.** The number of bytes of the output tensor of $v_i$ multiplied by a communication factor. In our case, we set the communication factor equal to 4. We benchmark the execution time of the simulator versus the real execution engine. We tried communication factor values from 1 to 10 and found 4 to be the closest match for the simulator with respect to the real execution engine.

## E.1  STATIC GRAPH FEATURES $X_{\mathcal{G}}$

The graph features matrix $X_{\mathcal{G}}$ is an $n \times 5$ matrix, where each row contains the following five features:

- **Computation cost for $v_j$.**
- **Sum of communication cost from predecessor nodes to $v_j$.** The sum of communication costs for all $e_{i,j}$ such that $(v_i, v_j) \in \mathcal{E}$.
- **Sum of communication cost from $v_j$ to descendant nodes.** The sum of communication costs for all $e_{j,k}$ such that $(v_j, v_k) \in \mathcal{E}$.
- **t-level cost of $v_j$.** The sum of all computation and communication costs on a t-level path for $v_j$. A t-level path is defined in Section 4.2.
- **b-level cost of $v_j$.** The sum of all computation and communication costs on a b-level path for $v_j$. A b-level path is defined in Section 4.2.

## E.2  DYNAMIC DEVICE FEATURES $X_{\mathcal{D}}$ FOR DEVICE $d$ AT TIME STEP $t$ GIVEN NODE $v$

The device features matrix $X_{\mathcal{D}}$ is a $|\mathcal{D}| \times 5$ matrix, where each row contains the following five features:

- **Total node computation cost.** The sum of the computation costs for all nodes that have been assigned to device $d$ at time step $t$.
- **Total predecessor node computation cost.** The sum of computation costs for all predecessor nodes of target node $v$ at time step $t$ that are currently assigned to device $d$.
- **Minimum start time of all inputs.** The earliest time to start executing a predecessor node of $v$ on device $d$ at time step $t$.
- **Maximum end time of all inputs.** The latest time for all predecessor nodes of $v$ to finish on device $d$ at time step $t$.
- **Earliest start time to compute $v$.** The earliest time for device $d$ to finish receiving inputs and start executing $v$.

# F  SYNCHRONOUS SYSTEM CONFIGURATION

In Table 1, we run CHAINMM using ScalaPack and FFNN using Pytorch Lightning with tensor parallel on 4 NVIDIA Tesla P100 GPUs with 16GB memory each.

## G   MORE ABLATION STUDIES

We conduct additional ablation studies on (1) the simulator implementation, (2) random seeds used in training the dual policy, (3) the number of message-passing rounds per episode, and (4) the inclusion of pre-training stages for PLACETO. We aim to evaluate the performance and training efficiency of our three-stage training approach with a dual-policy design compared with alternatives.

### G.1   SIMULATOR ABLATION STUDIES

We compare the simulated and real system execution times for the same device assignments in the following plots. **We hypothesize that the simulator serves as a cost-effective way to approximate system execution times without sacrificing significant precision.**

On the left of Fig. 26, we show the running times for both the simulator and the real system when training the dual policy on the `ChainMM` computation graph via imitation learning. We observe that the simulator tends to overestimate the running time compared to the real system and has some difficulty differentiating between assignments with similar running times. However, the simulator provides approximate running times that follow the same trend as the real system (e.g., high-quality assignments tend to have shorter running times, and low-quality ones exhibit longer running times).

On the right of Fig. 26, we present a statistical analysis comparing the performance of the simulator with the real system under the same setup. We find a Pearson coefficient of 0.79 and a Spearman coefficient of 0.69. This ablation study demonstrates that the simulator offers a cost-effective way to approximate system execution times.

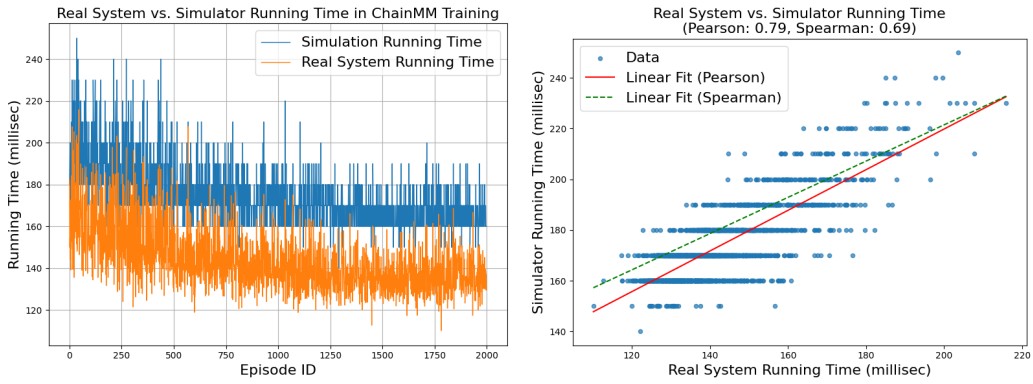

Figure 26: (left) A line chart showing a comparison between the simulator running time and the real system running time throughout training. (right) A scatter plot showing the simulator running time versus the real system running time with Pearson and Spearman fitting lines.

### G.2   EXPERIMENT WITH RANDOM SEEDS

In this study, we aim to test **the hypothesis that the best assignment found by our approach, DOPPLER, remains consistent across different random seeds during training**. Due to the cost of training, we are unable to run experiments multiple times with different random seeds for all computational graphs and methods. Therefore, we conduct five training runs of DOPPLER-SYS with different seeds on the `ChainMM` computation graph to test this hypothesis. This experimental setup—including the computation graph, dual-policy architectures, and training hyperparameters—is identical across runs, differing only in the random seeds. For each training run, we evaluate the best-found assignment over 10 system executions and report the mean and standard deviation in Table 5. The results show that DOPPLER achieves consistent performance across different random seeds.

| Model | Run1 | Run2 | Run3 | Run4 | Run5 |
|---|---|---|---|---|---|
| ChainMM | $123.2 \pm 3.7$ | $119.6 \pm 2.2$ | $122.7 \pm 2.1$ | $123.9 \pm 2.5$ | $121.7 \pm 0.9$ |

Table 5: Experiment running DOPPLER using different random seeds on CHAINMM computation graph. We test the best assignment found at the end of the training across different seeds with 10 system runs and report the mean and standard deviation of the system running time (in milliseconds) for each assignment.

### G.3 MESSAGE-PASSING ABLATION STUDIES

To enhance training efficiency for large computation graphs, we apply message passing on the graph once per MDP episode instead of once per MDP step. When message passing is applied once per MDP step, the number of message-passing rounds per episode equals the number of nodes in the graph. **We hypothesize that this modification has a negligible impact on DOPPLER's convergence but significantly reduces training time—proportional to the number of nodes in the computation graph.** We conduct this ablation study on the `ChainMM` computation graph using the simulator to save time, since per-step message passing incurs prohibitively high training costs.

| | DOPPLER-SIM | DOPPLER-SIM-mpnn-per-step |
|---|---|---|
| Best assignment | $122.5 \pm 4.0$ | $121.7 \pm 3.2$ |
| Best assignment @ Number of episodes | 3425 | 963 |
| Best assignment @ Number of message passing | 3425 | 107,856 |
| Run time reduction | 0.7% | |
| Extra message-passing | 3049.1% | |

Table 6: Running time (in milliseconds) for the best device assignment found at the end, along with the number of message-passing steps conducted until finding the best assignment. The reported time for the best assignment includes both the average and standard deviation of the system running time over 10 system rounds. DOPPLER-SIM refers to performing message passing on the computation graph per MDP episode, while DOPPLER-SIM-mpnn-per-step denotes conducting message-passing per MDP step within each episode.

In each MDP step within an episode, we assign a device to the currently selected node. Therefore, for the DOPPLER-SIM-mpnn-per-step approach, the number of message-passing rounds per episode equals the number of nodes in the computation graph. The `ChainMM` computation graph consists of 112 nodes. Table 6 shows that the best assignments–reported in the first row by their running times– were found at episode 3425 for DOPPLER-SIM and at episode 963 for DOPPLER-SIM-mpnn-per-step. Although the best assignment was found in fewer episodes with DOPPLER-SIM-mpnn-per-step, completing 963 episodes took significantly more wall-clock time than 3425 episodes for DOPPLER-SIM, because message-passing took the majority of time during training.

We evaluate the efficiency of the two approaches based on the number of message-passing rounds required to find the best assignment. DOPPLER-SIM performs 3425 message-passing operations (one per episode), while DOPPLER-SIM-mpnn-per-step conducts 963 episodes × 112 nodes = 107,856 message-passing operations (one per MDP step, or equivalently, per node in the computation graph). DOPPLER-SIM-mpnn-per-step achieves a 0.7% reduction in runtime for the best assignment compared to DOPPLER-SIM, but at the cost of 3049.1% more message-passing. Therefore, these results support our hypothesis that the modified approach has greatly reduced the training time with negligible impact on the performance.

### G.4 PLACETO ABLATION STUDIES

We conduct an ablation study on policy design for learning device assignments in both DOPPLER and PLACETO, explicitly including pre-training stages for PLACETO to isolate the effect of the

underlying policy design from the benefits of pre-training. **We hypothesize that the dual-policy design in DOPPLER outperforms PLACETO regardless of the inclusion of training stages.** To test our hypothesis, we pre-train the PLACETO policy using imitation learning and compare it with DOPPLER-SIM, which is trained using the imitation learning stage (Stage I) and the simulation-based RL stage (Stage II).

|  | PLACETO-pretrain | PLACETO | DOPPLER-SIM | DOPPLER-SYS |
|---|---|---|---|---|
| Best Assignment | $99.0 \pm 5.7$ | $126.3 \pm 5.8$ | $49.9 \pm 1.1$ | $47.4 \pm 0.7$ |

Table 7: The mean and standard deviation of the running time (in milliseconds) for the best assignment found for DOPPLER, compared to PLACETO and its pre-training version, PLACETO-pretrain, over 10 system runs. The results indicate that even with the pre-training stage, PLACETO-pretrain performs worse than DOPPLER-SIM.

Table 7 shows that DOPPLER discovers more effective device assignments than PLACETO. This result isolates the impact of the training stages and supports the claim that DOPPLER's dual-policy design outperforms the policy design in PLACETO.

## H    DOPPLER'S EXPERIMENTS ON DIFFERENT HARDWARE CONFIGURATIONS

The experiments so far demonstrate that DOPPLER outperforms the alternatives on four Tesla P100 GPUs, each with 16GB of memory. We hypothesize that DOPPLER can find better device assignments than alternatives across different hardware configurations. In this section, we conduct experiments with DOPPLER and other methods under (1) varying GPU memory sizes on four P100 GPUs and (2) different numbers and types of GPUs. Each experimental setup is described in the following two subsections.

### H.1    EXPERIMENTS WITH RESTRICTED GPU MEMORY

**We aim to test the hypothesis that DOPPLER can adapt to hardware setups with restricted GPU memory.** Table 8 shows results on four NVIDIA P100 GPUs, each restricted to use 8GB out of their 16GB total memory. DOPPLER learns to adapt to memory constraints, achieving up to 49.6% and 18.6% runtime reductions compared to the best baseline and ENUMERATIVEOPTIMIZER, respectively, while heuristics fail to adapt due to dynamic memory allocations in WC systems. These results confirm that DOPPLER can adapt to restricted memory settings.

|  |  | 4 GPUs | | | | RUNTIME REDUCTION | |
|---|---|---|---|---|---|---|---|
| MODEL | 1 GPU | CRITICAL PATH | PLACETO | ENUMOPT. | DOPPLER-SYS | BASELINE | ENUMOPT. |
| CHAINMM | $439.8 \pm 4.6$ | $310 \pm 4.9$ | $243.5 \pm 5.9$ | $133.5 \pm 10.4$ | $\mathbf{122.6} \pm 2.2$ | 49.6% | 8.2% |
| FFNN | $148.2 \pm 19.4$ | $225.8 \pm 19.4$ | $126.8 \pm 5.7$ | $49.2 \pm 0.9$ | $\mathbf{46.0} \pm 0.8$ | 63.7% | 6.5% |
| LLAMA-BLOCK | $465.1 \pm 7.8$ | $216.8 \pm 4.6$ | $433.5 \pm 6.2$ | $233.8 \pm 8.1$ | $\mathbf{190.2} \pm 11.2$ | 12.3% | 18.6% |
| LLAMA-LAYER | $482.6 \pm 9.4$ | $292.5 \pm 5.1$ | $302.1 \pm 20.2$ | $172.8 \pm 4.3$ | $\mathbf{154.0} \pm 3.7$ | 47.4% | 10.9% |

Table 8: Real engine execution times (in milliseconds) for assignments identified by our approaches (EnumerativeOptimizer, Doppler-SYS) using 4 GPUs with access to 8G out of 16G GPU memory for each GPU compared against those produced using 1 GPU and by two baselines (CRITICAL PATH and PLACETO). The results show that Doppler-SYS outperforms all baselines across all settings.

### H.2    EXPERIMENTS ON DIFFERENT NUMBERS AND TYPES OF GPUs

**We aim to test the hypothesis that DOPPLER can find better device assignments regardless of the number and type of GPUs in the server.** Table 9 presents results from running various computation graph architectures on eight NVIDIA V100 GPUs, each with 32GB of memory. The setup consists of two device meshes, each fully interconnected via NVLink, with a total of four additional NVLinks spanning between the meshes. We observed that DOPPLER effectively leverages NVLink to minimize

inter-mesh data transfer. We show that DOPPLER achieves up to a 67.7% runtime reduction compared to the existing baseline and up to 19.3% compared to ENUMERATIVEOPTIMIZER.

| MODEL | 1 GPU | 8 GPUs | | | RUNTIME REDUCTION | |
|---|---|---|---|---|---|---|
| | | CRITICAL PATH | ENUMOPT. | DOPPLER-SYS | BASELINE | ENUMOPT. |
| CHAINMM | $83.5 \pm 4.1$ | $69.6 \pm 2.6$ | $33.5 \pm 2.5$ | $\mathbf{32.1} \pm 0.7$ | 53.9% | 4.1% |
| FFNN | $51.4 \pm 1.8$ | $50.0 \pm 6.0$ | $20.0 \pm 2.6$ | $\mathbf{16.2} \pm 2.3$ | 67.7% | 19.3% |
| LLAMA-BLOCK | $154.4 \pm 6.3$ | $117.6 \pm 6.0$ | $\mathbf{109.6} \pm 4.2$ | $109.7 \pm 3.0$ | 6.7% | -0.1% |
| LLAMA-LAYER | $105.0 \pm 4.8$ | $105.4 \pm 4.2$ | $97.5 \pm 1.1$ | $\mathbf{90.6} \pm 4.1$ | 13.7% | 7.1% |

Table 9: Real engine execution times (in milliseconds) for assignments identified by our approaches (EnumerativeOptimizer, Doppler-SYS) using 8 V100 GPUs compared against those produced using 1 GPU and by one baseline (CRITICAL PATH). The results show that Doppler-SYS outperforms the alternatives in 3 out of 4 settings.

## I  DISCUSSION ON SCALING TO MUCH LARGER DATAFLOW GRAPHS

With respect to much larger graphs, we expect the linear scale-up to continue, but in practice, massive graphs are unlikely to be an issue. While we cannot know what companies such as OpenAI are doing, in practice, inference is likely not run on more than a few dozen GPUs (graphs grow linearly in size with increasing GPU counts). Training, while it requires thousands of GPUs, is performed by relatively independent machines through data parallelism and pipelining—each machine gets an identical transformer or MoE layer and a subset of the data—so it runs the same computation as all of the other machines. In this case, one would likely not train a single massive graph. Rather, each repeated block or layer would be used to train a separate dual policy in parallel and be given the same assignment on each machine with repeated structure throughout the cluster (assuming uniform hardware), with the runtimes collected across the cluster used to compute a reward.

## J  DOPPLER'S TRANSFER ABILITY ACROSS HARDWARE AND DETAILED ANALYSIS

We conducted a transfer learning study in which DOPPLER is trained on a computation graph running on four P100 GPUs with full NVLink connectivity (i.e., every GPU is directly connected to the others), and then evaluated on a system with eight V100 GPUs that have only partial NVLink connectivity. In the eight-GPU system, the V100 GPUs are organized into two groups of four (GPU 0–3 and GPU 4–7). Within each group, the GPUs are fully connected via NVLink. However, connectivity between the two groups is limited: only four NVLink connections link the groups in total. This creates a hierarchical topology with high-bandwidth communication within each 4-GPU group and more restricted bandwidth across groups.

In the assignment found by DOPPLER, we measured how the number of data transfers—and the ratios shown in parentheses—across the two 4-GPU groups, within the same 4-GPU group, and within the same single GPU change from zero-shot to 2K-shot for FFNN on eight GPUs, as reported in Table 10.

Table 11 shows the real engine execution times (in milliseconds) for the assignment identified by DOPPLER under different few-shot settings (Zero-shot, 2K-shot) compared against the baseline for the transfer learning study above. After 2K episodes, DOPPLER finds better assignments for both ChainMM and FFNN compared to training solely on eight GPUs without generalization (8K episodes) for runtime reduction of 19.0% (ChainMM) and 11.1% (FFNN).

| | ACROSS GROUPS | SAME GROUP | SAME GPU |
|---|---|---|---|
| ZERO-SHOT | 22 (10.6%) | 14 (6.7%) | 172 (82.7 %) |
| 2K EPISODES | 7 (3.4%) | 4 (1.9%) | 197 (94.7 %) |

Table 10: Data transfers across two 4-GPU groups, within the same 4-GPU group, and within the same single GPU changes from Zero-shot to 2K-shot for FFNN on eight GPUs.

| COMPUTE GRAPH | TRAIN HARDWARE | TARGET HARDWARE | ZERO-SHOT | 2K-SHOT | DOPPLER-SYS | CRIT. PATH | ENUM. OPT. |
|---|---|---|---|---|---|---|---|
| CHAINMM | 4 P100 GPUs | 8 V100 GPUs | 59.2 (1.9) | 26.0 (0.5) | 32.1 (0.7) | 69.6 (2.6) | 33.5 (2.5) |
| FFNN | 4 P100 GPUs | 8 V100 GPUs | 23.1 (2.3) | 14.4 (2.5) | 16.2 (2.3) | 50.0 (6.0) | 20.0 (2.6) |

Table 11: Real engine execution times (in milliseconds) for the assignment identified by DOPPLER under different few-shot settings (Zero-shot, 2K-shot) compared against the baseline.

