# OpenReview forum: "DOPPLER: Dual-Policy Learning for Device Assignment in Asynchronous Dataflow Graphs"
_ICLR.cc/2026/Conference — ICLR 2026 Poster_

### Official Review · Reviewer_navF · 2025-10-25

**Soundness:** 2
**Presentation:** 3
**Contribution:** 2
**Rating:** 4
**Confidence:** 4

**Summary:**

This paper shows DOPPLER, a dual-policy reinforcement learning framework for optimizing device assignment in asynchronous.
Traditional synchronous scheduling enforces global barriers and that leads to idle GPUs.
The presented method addresses device placement in an asynchronous dataflow environment where operations can be executed dynamically. The framework does a three-stage pipeline: (1) imitation learning using heuristic supervision, (2) reinforcement learning in simulation, and (3) online fine-tuning during real deployment. Experimental results on dataflow graphs from matrix chains, feedforward networks, and Llama layers show up to 52.7% reduction in execution time relative to state-of-the-art baselines. The method also generalizes to new hardware and architectures with few-shot adaptation.

**Strengths:**

The strengths are:

The paper is well written and organized. It also shows ablation studies and pseudocode.

The work addresses device placement under asynchronous execution where small per-query improvements scale to substantial cost reductions at deployment scale. A dual-policy decomposition separates operation selection and device placement. Then they introduce a three-stage training pipeline that bridges imitation learning, simulation-based RL, and real-system adaptation.

The framework’s shows transferability and few-shot adaptability to new hardware that is relevant for real-world heterogeneous distributed inference and training systems.

**Weaknesses:**

The weaknesses are:

The paper is missing explanation or studies for convergence or sample complexity analysis of the dual-policy setup. Adding those could strengthen the understanding of stability during online adaptation.

The experiments focus on up to 8 GPUs. It remains unclear how DOPPLER scales to large GPU clusters (256 GPUs) or heterogeneous systems with varying compute/memory characteristics.

The paper presents a 3 stage training, which seems rather complex. The incremental benefit of Stage III appears limited, and simpler two-stage alternatives could potentially achieve similar performance with reduced engineering overhead. The paper mentions a simulator used in Stage II but offers limited quantitative validation of how closely its performance correlates with real hardware results. This impacts the credibility of simulation-based pretraining.

**Questions:**

A few questions about the paper:

In the Stage III online phase, how is policy stability ensured when runtime-derived rewards could be noisy, is there a mechanism to prevent noisy updates?

What is the observed correlation between simulated runtime rewards and real execution times? This could validate the Stage II training phase and strengthen the paper

Would integrating model-parallel or operator-fusion heuristics into DOPPLER’s action space further improve placement quality?

**Details Of Ethics Concerns:**

-

---

> ### Author Response · Authors · 2025-11-21
> **Rebuttal Part 1**
>
> >**(W1.) “The paper is missing explanation or studies for convergence or sample complexity analysis of the dual-policy setup. Adding those could strengthen the understanding of stability during online adaptation. ”**
>
>
> Most convergence and sample-complexity analyses for policy-gradient methods assume a standard finite MDP with a deterministic reward function $R(s,a)$. This structure underlies the guarantees in classical policy-gradient theory—e.g., finite-time convergence of PG in discounted MDPs [1] and sample-complexity bounds for vanilla PG under smoothness and bounded-variance assumptions [2].
> DOPPLER does not satisfy these assumptions. For a given assignment A, the reward $R(A) = -ExecTime(A)$ is a random variable, not a deterministic function: execution time varies across runs due to asynchronous work-conserving scheduling, kernel overlap, and system contention in both the simulator and real hardware. Therefore, our objective is to minimize
>
> $$J(\theta)=\mathbb{E}_{\tau}[\mathbb{E}[R(A(\tau))]]$$
>
> where $ \tau \sim \pi_\theta$. The inner expectation is over a non-stationary distribution of execution times, violating the fixed-reward, fixed transition assumptions used in existing PG convergence/sample-complexity proofs.
> Because the theoretical conditions in prior work [1,2] do not hold in this stochastic Work Conserving-runtime setting, proving the same guarantees is not straightforward. Developing convergence and sample-complexity theory tailored to DOPPLER’s distributed execution environment is therefore left for future work.
>
> [1] Xiao, Lin. "On the convergence rates of policy gradient methods." Journal of Machine Learning Research 23.282 (2022): 1-36.
>
> [2] Yuan, Rui, Robert M. Gower, and Alessandro Lazaric. "A general sample complexity analysis of vanilla policy gradient." International Conference on Artificial Intelligence and Statistics. PMLR, 2022.
>
> >**(W2.) “The experiments focus on up to 8 GPUs. It remains unclear how DOPPLER scales to large GPU clusters (256 GPUs) or heterogeneous systems with varying compute/memory characteristics.”**
>
> With respect to larger systems, while we cannot know what companies such as OpenAI are doing, in practice, inference is probably not run on more than a few dozen GPUs (graphs grow linearly in size with increasing GPU counts). So this is not substantially more complicated than what we have done.
>
> Training, while it requires thousands of GPUs, is made by relatively independent access to the machines by data parallelism [1] and pipelining [2]—each machine gets an identical transformer/MoE layer [3] and a subset of the data [1] —so it is running the same computation as all of the other machines. In this case, each repeated block or layer would be used to train a separate, dual policy and given the same assignment on each machine throughout the cluster (assuming uniform hardware), with the runtimes collected across the cluster would be used to compute a reward. Thus, we argue that DOPPLER will be useful, even in a much, much larger context.
>
> [1] Li, Shen, et al. "Pytorch distributed: Experiences on accelerating data parallel training." arXiv preprint arXiv:2006.15704 (2020).
>
> [2] Narayanan, Deepak, et al. "Efficient large-scale language model training on gpu clusters using megatron-lm." Proceedings of the international conference for high performance computing, networking, storage and analysis. 2021.
>
> [3] Shi, Shaohuai, et al. "Schemoe: An extensible mixture-of-experts distributed training system with tasks scheduling." Proceedings of the Nineteenth European Conference on Computer Systems. 2024.

---

> > ### Author Response · Authors · 2025-11-21
> > **Rebuttal Part 2**
> >
> > >**(W3.) “The paper presents a 3 stage training, which seems rather complex. The incremental benefit of Stage III appears limited, and simpler two-stage alternatives could potentially achieve similar performance with reduced engineering overhead. The paper mentions a simulator used in Stage II but offers limited quantitative validation of how closely its performance correlates with real hardware results. This impacts the credibility of simulation-based pretraining.”**
> >
> > Appendix G.1 evaluates the simulator, which provides a cost-effective approximation of execution times. The need for Stage III stems from our observation that certain performance gains only surface during deployment, where real hardware effects—such as contention and communication variability—are not fully captured by simulators. These gaps widen with increasing graph complexity: FFNN shows similar performance under DOPPLER-SIM and DOPPLER-SYS, while Llama-Block differs by 19.46%. Stage III remains lightweight by design, and the inductive dual-policy architecture uses deployment rewards collected “for free” to adapt the policy with minimal overhead, effectively complementing simulator-based pretraining.
> >
> >
> > >**(Q1.) “In the Stage III online phase, how is policy stability ensured when runtime-derived rewards could be noisy, is there a mechanism to prevent noisy updates?”**
> >
> > During pre-training (Stages I and II), we use a larger learning rate (1e-3) and full exploration ($\epsilon$ = 1.0). In Stage III, we begin from a well-initialized policy and therefore switch to a smaller learning rate (1e-4) and a reduced exploration rate ($\epsilon$ = 0.2), which we gradually anneal to 0.0. This schedule stabilizes training under noisy real-system rewards and prevents large, unstable policy updates.
> >
> >
> > >**(Q2.) “What is the observed correlation between simulated runtime rewards and real execution times? This could validate the Stage II training phase and strengthen the paper”**
> >
> > We provided a detailed analysis on the simulator versus the real execution engine in Appendix G.1. We showed that the simulator serves as a cost-effective way to approximate execution times. We observe that the simulator tends to overestimate the running time and has some trouble differentiating between assignments with similar running times. However, the simulator provides approximate running times that follow the same trend as the real system.
> >
> > >**(Q3.) “Would integrating model-parallel or operator-fusion heuristics into DOPPLER’s action space further improve placement quality?”**
> >
> > The computation graphs we evaluate (FFNN, CHAINMM, LLaMA-block, LLaMA-layer) are constructed using an automated decomposition strategy similar to Alpa [1]. Such strategies generalize “model-parallel”, allowing a wide variety of automatically-generated hybrid parallelism strategies, and attempt to optimally decompose AI computation into fine-grained operations. DOPPLER operates directly on the resulting operator graph.
> >
> > Operator-fusion heuristics, as explored in prior work [2], primarily aim to improve efficiency by modifying the computation graph before placement. This is orthogonal to our setting: DOPPLER assumes a given computation graph and optimizes device assignments for that graph. In practice, fused graphs would simply be provided as input to DOPPLER, so fusion and learned device placement can be composed.
> >
> > [1] Zheng, Lianmin, et al. "Alpa: Automating inter-and {Intra-Operator} parallelism for distributed deep learning." 16th USENIX Symposium on Operating Systems Design and Implementation (OSDI 22). 2022.
> >
> > [2] Mirhoseini, Azalia, et al. "Device placement optimization with reinforcement learning." International conference on machine learning. PMLR, 2017.

---

> > > ### Comment · Reviewer_navF · 2025-11-23
> > > **rebuttal acknowledgement**
> > >
> > > Thank you for authors clarification and address of the weaknesses.

---

### Official Review · Reviewer_hjpH · 2025-10-25

**Soundness:** 3
**Presentation:** 2
**Contribution:** 2
**Rating:** 4
**Confidence:** 3

**Summary:**

### **Review Summary**

This paper proposes DOPPLER, a reinforcement learning (RL) framework for the device placement of dataflow graphs in an asynchronous, "work-conserving" (WC) system. The authors argue that existing bulk-synchronous systems under-utilise resources and that prior RL solutions use a single, inflexible placement policy. DOPPLER's core contributions are a **dual-policy architecture**—a selection policy (SEL) to choose the next node and a placement policy (PLC) to assign it—and a **three-stage training process** (Imitation Learning, RL in simulation, and RL on the real system). The method is shown to outperform heuristic and RL baselines in execution time across several ML workloads.

While the problem is highly relevant and the empirical results are strong, the paper's core contributions are not well-justified. The necessity of the dual-policy (SEL+PLC) over a single placement policy is poorly motivated and its benefits are conflated with other implementation choices. Furthermore, the 3-stage training pipeline relies on an online, real-world RL stage, which is impractical for the very production environments the paper targets, undermining the "cost-effective" claims.

**Strengths:**

### **Strengths**

1.  **Important Problem:** The paper correctly identifies a key inefficiency in modern ML systems: bulk-synchronous execution (e.g., all-reduce) leads to device idle time [cite: 69-72]. It convincingly argues that asynchronous, work-conserving (WC) systems offer significant speedups (Table 1) [cite: 161-163], but that optimizing for them is a more complex temporal problem [cite: 169-171].

2.  **Comprehensive Training Framework:** The three-stage training paradigm (Stage I: Imitation Learning, Stage II: Simulation-based RL, Stage III: Real-system RL) is a thorough approach [cite: 177-180, 487]. Figure 3 demonstrates that this curriculum is highly effective, using imitation and simulation for a strong warm start before the final, high-performance real-system tuning [cite: 731-734].

3.  **Strong Empirical Performance:** The final model, DOPPLER-SYS (which uses all 3 stages), demonstrates state-of-the-art performance. It achieves significant execution time reductions (up to 52.7%) compared to all baselines, including prior RL methods (PLACETO, GDP) and a strong, purpose-built heuristic (ENUMERATIVE OPTIMIZER) [cite: 619-621, 642-644].

4.  **Pragmatic Scalability:** The paper includes a critical implementation detail: running GNN message passing only *once per episode* (per graph) rather than *once per step* (per node) [cite: 481-485]. Appendix G.3 correctly identifies this as a massive efficiency gain (a 30x reduction in message-passing operations for a <1% performance trade-off), making the approach scalable and computationally viable [cite: 2331-2334, 2399-2400].

**Weaknesses:**

### **Weaknesses and Questions**

1.  **Weak Motivation for Dual Policy:** The paper's central contribution is the "dual-policy" (SEL+PLC) architecture [cite: 61, 174-176, 182]. However, the *reason* for factoring the policy this way is poorly justified. The paper never compares this two-policy agent against the most obvious and standard baseline: a *single* policy that directly outputs a `(node, device)` pair. The ablation in Table 3 [cite: 646-649] only compares DOPPLER to variants where one of the two policies is replaced by a heuristic; this does not prove that the dual-policy factorization itself is beneficial.

2.  **Misleading Scalability Claims:** The paper's impressive scaling results (Fig. 4) [cite: 737-739] are presented as a benefit of DOPPLER. However, the text (and Appendix G.3) reveals this speedup is due to an implementation choice (GNN-per-episode) that is *orthogonal* to the dual-policy architecture [cite: 481-485, 2331-2400]. The paper criticizes PLACETO for being "prohibitive" and "inefficient" because it runs the GNN per-step [cite: 482-483], but this is an apples-to-oranges comparison. A fair comparison would require re-implementing all baselines with the same "GNN-per-episode" optimization. As such, the scalability results do not support the *dual-policy architecture* itself.

3.  **Impracticality of Stage III (Real System RL):** The paper's best results rely on `DOPPLER-SYS`, which uses Stage III, an online RL stage on the "real hardware" [cite: 180, 552-554]. The paper claims this is "for free"[cite: 554], which is highly misleading. This stage requires deploying an *exploratory* (i.e., non-optimal) policy in a production environment, forcing users to suffer through the high-variance, low-performance assignments generated during exploration. Figure 3 ("0k-0k-8k" curve) shows exactly how slow and unstable this "from-scratch" online learning is [cite: 732-734]. This reliance on an impractical training step undermines the paper's applicability.

4.  **Simulator-Reality Gap:** The only reason the impractical Stage III is necessary is that the simulator in Stage II is not accurate enough. This is shown by the consistent performance gap between `DOPPLER-SIM` (sim-trained) and `DOPPLER-SYS` (real-trained) in Table 2[cite: 619, 643]. The paper explicitly confirms this gap in Appendix G.1, stating the simulator "tends to overestimate" and has "trouble differentiating between assignments" [cite: 2220-2221]. The 3-stage framework is thus not just a feature but a *crutch* to overcome a flawed simulator.

**Questions:**

none

---

> ### Author Response · Authors · 2025-11-21
> **Rebuttal Part 1**
>
> >**(W1.) “Weak Motivation for Dual Policy: The paper's central contribution is the "dual-policy" (SEL+PLC) architecture [cite: 61, 174-176, 182]. However, the reason for factoring the policy this way is poorly justified. The paper never compares this two-policy agent against the most obvious and standard baseline: a single policy that directly outputs a (node, device) pair...”**
>
> To directly address the reviewer’s concern, we conducted an additional ablation in which we replaced the dual-policy architecture with a single policy that outputs a joint (node, device) action. For a fair comparison, we preserved the same feature sets—node features and device features—and constructed joint features representing all possible (node, device) combinations. The single-policy agent was trained using the same three-stage procedure to ensure a fair evaluation.
>
> The table below reports the mean and standard deviation of execution times over ten system runs for the best assignment found by each method on both ChainMM and FFNN graphs, along with the episode index at which that best assignment was discovered.:
>
> ||ChainMM|FFNN|
> |:---:|:--:|:--:|
> |Doppler Single Policy Execution Time|132.0 (2.6)|49.2 (0.8)
> |Doppler Dual-policy Execution Time |**123.4** (2.5)|**47.4** (0.7)|
> |Doppler Single Policy episode id for the best assignment|2975|3027|
> |Doppler Dual-policy episode id for the best assignment|**1340**|**1882**|
> |Performance drop| 3.7%|7.0%|
>
> The dual-policy architecture finds high-quality assignments much earlier: the best assignments appear at episode 1340 (FFNN) and 1882 (ChainMM) for dual-policy, compared to episodes 3027 (FFNN) and 2975 (ChainMM) for the single-policy agent. Moreover, we observe that merging the two policies into a single policy leads to a 7.0% degradation on ChainMM and 3.7% degradation on FFNN, despite using identical features and training procedure. This performance drop stems from the dramatically larger action space of the single-policy agent (# of nodes × # of devices), which increases learning difficulty. This demonstrates that separating the “which node?” and “which device?” decisions into two coordinated policies improves both performance and sample efficiency.
>
> >**(W2.) “Misleading Scalability Claims: The paper's impressive scaling results (Fig. 4) [cite: 737-739] are presented as a benefit of DOPPLER. However, the text (and Appendix G.3) reveals this speedup is due to an implementation choice (GNN-per-episode) that is orthogonal to the dual-policy architecture [cite: 481-485, 2331-2400]. The paper criticizes PLACETO for being "prohibitive" and "inefficient" because it runs the GNN per-step [cite: 482-483], but this is an apples-to-oranges comparison. A fair comparison would require re-implementing all baselines with the same "GNN-per-episode" optimization. As such, the scalability results do not support the dual-policy architecture itself.”**
>
> First of all, we want to emphasize that the use of a single message passing step per episode is one of our contributions for improving efficiency with negligible performance drop due to dual-policy design, which we compute device features separated based on static graph embedding, while Placeto requires message passing after each action to propagate the device assignments from the previous step. We provided an ablation study showing that this one message passing design has a negligible effect on our performance in Appendix G.3.
>
> To directly address the reviewer’s concern, we conduct an experiment of running Placeto with one message passing per episode and report the mean and standard deviation in parentheses of real system execution time on the best assignment found over 10 systems run below, along with Doppler on ChainMM graph:
>
> ||per-step mpnn|per-episode mpnn|performance drop|
> |:---:|:--:|:--:|:--:|
> |Placeto|137.1 (2.2)|189.8 (7.7)|38.4%|
> |Doppler|122.5 (4.0)|121.7 (3.2)|0.7%|
>
> The results in the table show a 38.4% performance drop by changing per-step message passing to per-episode message passing on Placeto, while in Appendix G.3, we show that Doppler only faces a 0.7% performance drop. With the original Placeto setup, it has worse performance compared to Doppler (11.92%), and with per-episode MPNN it is more efficient but has 55.96% worse performance compared to DOPPLER.
>
> For the other learning-based baseline GDP, it is outputting device assignment in one shot without multiple message passing within an episode. However, it has a placement policy network with GraphSAGE as the GNN, combined with a transformer architecture policy network with a recurrence mechanism, which is more expensive compared to DOPPLER.

---

> > ### Author Response · Authors · 2025-11-21
> > **Rebuttal Part 2**
> >
> > >**(W3.) “Impracticality of Stage III (Real System RL): The paper's best results rely on DOPPLER-SYS, which uses Stage III, an online RL stage on the "real hardware" [cite: 180, 552-554]. The paper claims this is "for free"[cite: 554], which is highly misleading. This stage requires deploying an exploratory (i.e., non-optimal) policy in a production environment, forcing users to suffer through the high-variance, low-performance assignments generated during exploration. Figure 3 ("0k-0k-8k" curve) shows exactly how slow and unstable this "from-scratch" online learning is [cite: 732-734]. This reliance on an impractical training step undermines the paper's applicability.”**
> >
> > We do not recommend the 0k–0k–8k setting; it is included only as an ablation. In fact, DOPPLER-SYS uses all three stages, and Stages I and II are specifically designed to avoid exposing users to the high-variance, low-performance exploration behavior that concerns the reviewer.  When entering Stage III, users could expect a reasonably good policy (as demonstrated by DOPPLER-SIM, which uses Stage I and II) and a further, gradually improving policy through Stage III training to further improve performance.
> >
> > Our use of “for free” refers to the fact that reward signals come directly from normal executions already occurring while serving user requests in production, requiring no additional runs to collect reward signals for RL training in Stage III. Figure 3 illustrates the benefits of combining all three stages (eg, 2k-8k-8k) for training, with a low starting run time compared with the alternatives and further reduced running time with stage III training, achieved by running on a real system. We made corresponding changes on lines 384 - 388 for clarification regarding DOPPLER-SYS and DOPPLER-SIM.
> >
> >
> > >**(W4.) “Simulator-Reality Gap: The only reason the impractical Stage III is necessary is that the simulator in Stage II is not accurate enough. This is shown by the consistent performance gap between DOPPLER-SIM (sim-trained) and DOPPLER-SYS (real-trained) in Table 2[cite: 619, 643]. The paper explicitly confirms this gap in Appendix G.1, stating the simulator "tends to overestimate" and has "trouble differentiating between assignments" [cite: 2220-2221]. The 3-stage framework is thus not just a feature but a crutch to overcome a flawed simulator.”**
> >
> > DOPPLER-SIM corresponds to Stage I + Stage II only (imitation learning + RL based on the simulator), while DOPPLER-SYS corresponds to all three stages. Therefore, this is not a direct comparison between a simulator and a real system, but evidence supporting the Stage III training, which is intentionally designed as an online adaptation phase that tunes the policy to the actual hardware, interconnect topology, and workload mix of the real deployment environment.
> >
> > This three-stage pipeline is analogous to sim-to-real RL in RL-based compiler/hardware scheduling. No simulator can perfectly model dynamic GPU contention,  NVLink jitter, and system-level scheduling noise in a work-conserving multi-GPU environment. A detailed study on the simulator is provided in Appendix G.1, showing that the simulator serves as a cost-effective way to approximate system execution times without sacrificing significant precision to avoid high exploration costs during Stage III.
> >
> > Stage III is therefore not a crutch, but a principled and lightweight final step that allows DOPPLER‐SYS to adapt to the deployment system in a few episodes of online learning. The observed performance gap between DOPPLER-SIM and DOPPLER-SYS is expected and consistent with sim-to-real RL: the simulator provides a strong initialization, and Stage III refines the policy using real measurements.
> > We include this content on lines 315 - 323.

---

> > > ### Comment · Reviewer_hjpH · 2025-11-26
> > >
> > > Thank you for the clarification and addressed weaknesses

---

### Official Review · Reviewer_kPCd · 2025-10-30

**Soundness:** 3
**Presentation:** 3
**Contribution:** 3
**Rating:** 8
**Confidence:** 2

**Summary:**

The authors introduce DOPPLER, a method to efficiently assign computational operations in dataflow graphs to GPU devices in work-conserving (asynchronous) systems. The key innovation is a dual-policy learning framework that separates the problem into selecting which operation to assign next and determining which device should execute it. DOPPLER employs a three-stage training approach combining imitation learning, simulation-based RL, and continuous real-system RL during deployment, to achieve significant execution time reductions over existing baselines using clever design choices to ensure both good performance and scalability.

**Strengths:**

- Very significant practical impact in performance gains over best baselines towards the dataflow graph assignment problem.
- Well clever design choices to enable both scalable training and improved performance
	- Pretraining techniques to improve performance during the actual RL phase.
	- Clever message passing implementation to reduce training time with very negligible performance impact
- Strong generalization across hardware architectures

**Weaknesses:**

- Evaluation seems limited to relatively small graphs (~200 nodes). Real life workloads could be much bigger that could affect the linear scalability claim. In my own personal experience, scheduling larger graphs sometimes introduces additional complexity that sometimes is not captured on smaller graphs
- Dynamic Dataflow - I could imagine that dataflow may change dynamically during execution - especially with larger computational graphs. Is this a setting your algorithm can handle because from the writing this doesn't seem to be considered.

**Questions:**

- How does this work relate to the substantial body of research on learning-augmented scheduling? While your setting differs (multi GPU dataflow graphs vs. traditional job scheduling), the core problem structure appears similar:
	- Both involve sequential assignment decisions under uncertainty
	- Both use predictions/learning to schedule DAGs
	- Both must balance load and minimize communication/movement costs
	The learning augmented scheduling literature has developed concepts like consistency (performance with accurate predictions), robustness (worst case guarantees), and prediction error bounds. Could these frameworks provide theoretical grounding for DOPPLER? For example, can you characterize DOPPLER's consistency and robustness formally, or provide approximation guarantees based on simulator accuracy?
- How does your work scale to larger graphs? Is this a realistic setting for dataflow? And does your linear scalability still hold?
- How does this work for dynamic graphs (that is when the graph changes dynamically) - is this a possible setting for your problem?
- Why is an RL method needed for this problem versus approaches like Bayesian Optimization or other direct optimization methods?
- Hyperparameter selection? To deploy your algorithm in practice for heterogeneous workloads and hardware, how should one set hyperparameters?
- Why those specific neural network architectures? This is not a criticism - rather I'm seeking to understand why you made those design choices.

If some of my more important questions are addressed, I'm very inclined to increase my score. I think this will be a very impactful addition to the community speed up compute especially for ML jobs.

---

> ### Author Response · Authors · 2025-11-21
> **Rebuttal Part 1**
>
> We thank the reviewer for recognizing our contributions and for the opportunity to provide additional experiments and clarifications about DOPPLER’s scalability and comparisons to other approaches and settings. The content from the rebuttal will greatly strengthen our paper.
>
>
> >**(Q1.)“How does this work relate to the substantial body of research on learning-augmented scheduling? While your setting differs (multi GPU dataflow graphs vs. traditional job scheduling), the core problem structure appears similar…”**
>
> Doppler is a Learning-Augmented Algorithm (LAA), as it applies neural policies for the select and place steps. However, the main contribution of LAAs as a framework is enabling the theoretical analysis (i.e., approximation guarantees) for the quality and running time of the resulting algorithms based on properties of the classical algorithm being augmented and the predictor. The reason why we did not discuss Doppler in the context of LAAs is that (1) the classical heuristic Doppler augments (select+place) does not provide any known guarantees, and (2) it is hard to reason about the error of select and place policies, as such errors depend on the final (combinatorial) outcome.  This outcome, produced by executing on a stochastic real system, introduces additional complexity beyond the theoretical analysis.  We note that all the RL-based solutions we considered as baselines fall under the same case and do not provide any theoretical guarantees. Similarly, our paper focuses on validating our approach experimentally.
>
>
> >**“(Q2.) How does your work scale to larger graphs? Is this a realistic setting for dataflow? And does your linear scalability still hold?”**
>
> We address scalability to much larger dataflow graphs in Appendix J. Although we cannot know the exact configurations used by organizations such as OpenAI, inference workloads in practice rarely span more than a few dozen GPUs. Because dataflow graphs grow roughly linearly with GPU count, these inference graphs are not substantially more complex than the ones we evaluate.
> Training workloads do employ thousands of GPUs, but modern large-scale training relies heavily on data parallelism [1] and pipelining [2]. Each worker hosts an identical transformer/MoE layer [3] and processes a disjoint shard of the data [1], resulting in many repeated, structurally identical small subgraphs rather than one massive heterogeneous graph. In this regime, each repeated block or layer can share the same dual-policy assignment (assuming uniform hardware), and runtime measurements aggregated across workers can be used to compute the reward.
>
> For these reasons, the complexity of scheduling real-world large-scale workloads is dominated by many copies of small subgraphs—not by a single monolithic graph—and DOPPLER’s design naturally extends to this much larger context. Under this structure, DOPPLER’s linear scalability and architectural assumptions continue to hold, making it applicable even in substantially larger deployments.
>
> [1] Li, Shen, et al. "Pytorch distributed: Experiences on accelerating data parallel training." arXiv preprint arXiv:2006.15704 (2020).
>
> [2] Narayanan, Deepak, et al. "Efficient large-scale language model training on gpu clusters using megatron-lm." Proceedings of the international conference for high performance computing, networking, storage and analysis. 2021.
>
> [3] Shi, Shaohuai, et al. "Schemoe: An extensible mixture-of-experts distributed training system with tasks scheduling." Proceedings of the Nineteenth European Conference on Computer Systems. 2024.
>
> >**(Q3.)“How does this work for dynamic graphs (that is when the graph changes dynamically) - is this a possible setting for your problem?”**
>
> There is nothing in theory that prevents us from using our methods in a dynamic setting. All of our algorithms would work in a dynamic setting, though extensive on-the-fly changes to the graph may require the RL algorithm to learn to adapt to the new graph.
>
> That said, most modern applications to AI (including transformers and MoEs, both during training and inference) in practice use a static graph, and hence this was our focus. Even in the case of an MoE where dataflow/execution is dynamic, the computation is naturally modularized into per-expert static subgraphs, and the only dynamic component is the data-dependent selection of which expert subgraph is executed for a given time [1]. Thus, the overall dataflow graph is a fixed super-graph whose expert modules are selected dynamically. In this setting, DOPPLER applies directly: the SEL/PLC policies assign devices for each static module, and dynamic routing simply determines which modules run during each step without modifying graph topology.
>
> [1] Shi, Shaohuai, et al. "Schemoe: An extensible mixture-of-experts distributed training system with tasks scheduling." Proceedings of the Nineteenth European Conference on Computer Systems. 2024.

---

> > ### Author Response · Authors · 2025-11-21
> > **Rebuttal Part 2**
> >
> > >**(Q4.)“Why is an RL method needed for this problem versus approaches like Bayesian Optimization or other direct optimization methods?”**
> >
> > Bayesian Optimization (BO) is suitable for problems with a small number of parameters, and the goal is to minimize costly evaluations of a black-box objective function [1]. On the other hand, our problem has a combinatorial parameter space (of all possible assignments). In our problem, we address the evaluation cost using imitation learning and the simulator before optimizing based on the real system. Moreover, we use reinforcement learning to efficiently search over the combinatorial space of possible assignments in a sequential manner (i.e., assigning one node at a time). Some approaches combine BO and RL to improve sample efficiency [2].
> >
> > [1] Frazier. A tutorial on Bayesian optimization. 2018.
> >
> > [2] Brochu et al. A tutorial on Bayesian optimization of expensive cost functions, with application to active user modeling and hierarchical reinforcement learning. 2009.
> >
> > >**(Q5.) “Hyperparameter selection? To deploy your algorithm in practice for heterogeneous workloads and hardware, how should one set hyperparameters?”**
> >
> > Across all experiments—spanning multiple computation graphs and heterogeneous hardware—DOPPLER uses a single fixed set of hyperparameters (e.g., learning rate = 1e-4, exploration rate = 0.5). The fact that these same settings work well in all reported scenarios indicates that DOPPLER is not highly sensitive to hyperparameter choices and does not require careful hyperparameter search to achieve good performance.
> >
> > To further support this point, we provide additional results on FFNN graphs using several alternative hyperparameter configurations. As shown in the table below, the performance remains stable: for each configuration, we report the mean and standard deviation in parentheses of ten system execution times obtained using the best assignment found by DOPPLER below:
> >
> > |Learning Rate|3e-4|1e-4|5e-5|3e-5|
> > |:---:|:--:|:--:|:--:|:--:|
> > |Execution Time|45.7 (0.5)|47.4 (0.7)|47.0 (0.7)|46.2 (0.8)|
> >
> > |Exploration Rate|0.3|0.25|0.2|0.1|
> > |:---:|:--:|:--:|:--:|:--:|
> > |Execution Time|48.3 (0.8)|45.6 (0.5)|47.4 (0.7)|46.2 (0.9)|
> >
> > Across hyperparameter settings, the average execution times vary only modestly (45.6–48.3) and show very small standard deviations over 10 runs (0.5–0.9). This indicates that DOPPLER is robust to hyperparameter choices. In practice, users can simply adopt our recommended values or tune them with standard methods such as line search.
> >
> > >**(Q6.) “Why those specific neural network architectures? This is not a criticism - rather I'm seeking to understand why you made those design choices.”**
> >
> > Regarding the design of dual-policy neural network architectures, device assignment requires reasoning over dataflow graphs whose behavior is shaped by dependencies, critical paths, and communication edges. Graph neural networks naturally provide the right inductive bias for this setting: they propagate information along dependency edges, capture local structure, and generalize across graphs of varying sizes. The dual-policy decomposition is motivated by heterogeneity in both the workload and hardware, since each scheduling decision depends jointly on the node being assigned and the device on which it may execute. This modular design reflects the semantics of the dynamic environment and allows the same architecture to adapt across different workloads (CHAINMM, FFNN, LLAMA-BLOCK, LLAMA-LAYER) and across different hardware configurations. We incorporate these explanations on lines 237-243.
> >
> > For computation graph neural network architectures, we selected FFNN, CHAINMM, LLaMA-block, and LLaMA-layer because they represent the fundamental computational primitives underlying modern large-scale deep learning systems. FFNNs and chained matrix multiplications capture the dense linear-algebra–dominated workloads common across classical neural networks [1], while transformer blocks (e.g., LLaMA layers) represent the dominant architecture in contemporary large language models [2]. These components form the bedrock of today’s AI workloads, so evaluating DOPPLER on them ensures relevance to real production systems. We add this rationale on lines 349-351.
> >
> > [1] Markidis, Stefano, et al. "Nvidia tensor core programmability, performance & precision." 2018 IEEE international parallel and distributed processing symposium workshops (IPDPSW). IEEE, 2018.
> >
> > [2] Vaswani, Ashish, et al. "Attention is all you need." Advances in neural information processing systems 30 (2017).

---

> ### Comment · Reviewer_kPCd · 2025-11-25
> **Response to Authors**
>
> Thank you to the authors for the detailed feedback. I think it may be a good idea to include answers to Q1, Q2, and Q3 to the paper somewhere. It may help readability for those in an adjacent field. However, I maintain my score of 8.

---

> > ### Author Response · Authors · 2025-12-03
> >
> > Thank you for your thoughtful response. We have added our answers to Q1, Q2, and Q3 in the revised manuscript, highlighted in red at lines 482–485 (Q1), 453–456 (Q2), and 105–107 (Q3).

---

### Official Review · Reviewer_qMoz · 2025-11-02

**Soundness:** 1
**Presentation:** 1
**Contribution:** 2
**Rating:** 2
**Confidence:** 4

**Summary:**

This study developed a reinforcement learning based approach (DOPPLER) to enable efficient assign operations in a dataflow graph. Experiments shows that the trained RL model could reduce the execution time in neural network operations with different network architectures.

**Strengths:**

The problem has a clear motivation, illustrated by a dataflow graph, in the introduction section for matrix multiplication in deep learning operations.

**Weaknesses:**

#1 It is not easy for readers to follow the logic of this paper. The paper fails to clearly describe the problem and methods.

#2 The abstract is too short to demonstrate the significance of DOPPLER. Basically, there are no quantitative results in abstract

#3 “Simulator” for system execution was used through the paper without any description. Please clarify.

#4 Since section 3 was titled as problem definition, is section 2 just a background? More specifically, is Algorithm 1 an existing algorithm or the problem you aim to study in this paper? No citations were provided in Section 2.

#5 The paper does not provide sufficient technical details but rather forced everything to the appendix.

#6 GNN is too broad. What specific architecture did you use? Clarification and citation are needed in the main text.

#7 The rationale of three-stage training was not well justified. Any challenges or preliminary experiments that could support your proposed training mechanism?

#8 What reinforcement learning algorithm did you use?

#9 How does “RL for combinatorial optimization” in the related work section contribute to underpin the background of your work? E.g., you mentioned many works for TSP but does it help here?

#10 The “combinatorial structure” in line 197 was not clearly introduced in the main text.

**Questions:**

Please see weakness section.

---

> ### Author Response · Authors · 2025-11-21
> **Rebuttal Part 1**
>
> We thank the reviewer for the comments and the opportunity to clarify some of the ideas in our paper. All questions have been addressed, and the corresponding revisions have been incorporated into the paper.
>
> >**(W1.) “#1 It is not easy for readers to follow the logic of this paper. The paper fails to clearly describe the problem and methods.”**
>
> We expanded technical details in the main text, and use the additional page in the final version to incorporate further explanations, along with a summary index at the start of the appendix (lines 702–730). We strengthened the abstract by adding quantitative results (lines 20–21) and clarified the definition of the simulator (lines 178–179). To resolve confusion between background and problem definition, we retitled Sections 2 and 3 to clearly distinguish the device assignment problem from the reinforcement-learning formulation (lines 102, 182). We explained how prior work on RL for combinatorial optimization supports our approach (lines 472–474).
>
> > **(W2.) “#2 The abstract is too short to demonstrate the significance of DOPPLER. Basically, there are no quantitative results in abstract”**
>
> We have revised the abstract to include quantitative results, now shown on lines 20–21.
>
> >**(W3.)  “#3 “Simulator” for system execution was used through the paper without any description. Please clarify. ”**
>
> We describe a simulator as an engine for estimating execution time through simulating the real system’s event-driven behavior (described in Algorithm 1), including both kernel executions, inter-device communications, etc. We made the corresponding changes on lines 178-179.
>
> >**(W4.)  “#4 Since section 3 was titled as problem definition, is section 2 just a background? More specifically, is Algorithm 1 an existing algorithm or the problem you aim to study in this paper? No citations were provided in Section 2”**
>
> In this paper, we study the device assignment problem in the context of a work-conserving system, as described in Section 2. Section 3 then presents our solution framework based on reinforcement learning and explains how the problem is modeled. For clarity, we have updated the titles of Sections 2 and 3 (lines 102 and 182). Algorithm 1 specifies how a given assignment A executes under a work-conserving system. This execution procedure defines the objective whose total runtime we aim to minimize.

---

> > ### Author Response · Authors · 2025-11-21
> > **Rebuttal Part 2**
> >
> > >**(W5.)  “#5 The paper does not provide sufficient technical details but rather forced everything to the appendix.”**
> >
> > Our original submission placed several technical components in the appendix in order to comply with the conference’s strict page limits while keeping the main text focused on the core ideas. We will use the additional page allowed in the final version to incorporate your suggestions. To further improve readability while respecting the page limit, we add a concise summary list at the beginning of the appendix that guides readers to additional technical details and extended experiments on lines 702 - 730.
> >
> > >**(W6.)  “#6 GNN is too broad. What specific architecture did you use? Clarification and citation are needed in the main text.”**
> >
> > We adopted message passing neural networks with citation provided on line 240 (original submission). Details regarding the structures of the two GNNs are provided in paragraph with title “Node policy network” on lines 255 - 269 (original submission) and “Device policy network” on lines 270 - 280 (original submission).
> >
> > >**(W7.)  “#7 The rationale of three-stage training was not well justified. Any challenges or preliminary experiments that could support your proposed training mechanism?”**
> >
> > In Table 2, we report results for two variants of DOPPLER: DOPPLER-SIM (Stages I + II) and DOPPLER-SYS (all three stages). Both variants outperform all baselines, and DOPPLER-SYS—which incorporates the full three-stage framework—achieves the best performance on most tasks, highlighting the benefit of the complete training pipeline. In addition, Figure 3 provides a further analysis of different stage combinations and training-episode allocations. These results consistently show that the full three-stage version of DOPPLER yields the lowest execution time.
> >
> > >**(W8.)  “#8 What reinforcement learning algorithm did you use?”**
> >
> > On line 306 (original submission), we specify that our method employs a policy gradient–based reinforcement learning algorithm and include the corresponding citation.
> >
> > >**(W9.)  “#9 How does “RL for combinatorial optimization” in the related work section contribute to underpin the background of your work? E.g., you mentioned many works for TSP but does it help here?”**
> >
> > Similar to [1] and [2], we cite the “RL for combinatorial optimization” literature to provide conceptual background for why reinforcement learning is effective for problems with extremely large discrete action spaces, including device assignment. Although tasks such as TSP differ from our setting, these works establish two key principles that directly motivate DOPPLER: (1) RL can learn constructive, sequential heuristics that outperform hand-crafted solutions for combinatorial problems, and (2) RL methods can exploit structural similarity between related instances (e.g., different dataflow graphs), enabling generalization across problem sizes and topologies. Our approach adopts the same paradigm—learning a sequential decision process over a combinatorial space—while adapting it to the unique constraints of asynchronous GPU execution. We have revised this section to clarify the point on lines 472-474.
> >
> > [1] Paliwal, Aditya, et al. "Reinforced Genetic Algorithm Learning for Optimizing Computation Graphs." ICLR’20.
> >
> > [2] Mirhoseini, Azalia, et al. "Device placement optimization with reinforcement learning." ICML’17.
> >
> > >**(W10.)  “#10 The “combinatorial structure” in line 197 was not clearly introduced in the main text.”**
> >
> > The device assignment problem aims to map each computation (i.e., each node in the dataflow graph) to one of the available devices. Consequently, the solution space consists of all possible (node, device) assignment combinations, since any node may be placed on any device. For the example in lines 193–195 (original submission), we describe how a dataflow graph with 100 nodes and 8 GPUs yields a total of $8^{100}$=$2^{300}$ possible assignments.

---

> > > ### Comment · Reviewer_qMoz · 2025-11-25
> > > **Rebuttal acknowledgement**
> > >
> > > Thank you for the clarification and further updated the manuscript.

---

> > > > ### Author Response · Authors · 2025-11-27
> > > >
> > > > Thank you for taking the time to review our paper and for your engagement during the rebuttal process. We have addressed all concerns raised during the discussion phase, and we are glad to provide further clarification or additional experiments before the discussion period concludes. If our clarifications resolve your concerns, we would appreciate it if you would reconsider the current score.

---

### Meta-Review · Area_Chair_vYaG · 2026-01-06

**Summary:**

This paper proposes DOPPLER, a reinforcement learning (RL) framework for device placement in dataflow graphs under asynchronous, "work-conserving" systems. DOPPLER's core contributions include a dual-policy architecture that separates the problem into selecting which operation to assign next and determining which device should execute it. DOPPLER employs a three-stage training process (imitation learning, RL in simulation, and continuous real-system RL during deployment). The method is shown to outperform heuristic and RL baselines in execution time across several ML workloads. All four reviewers agree that the studied problem is important and that the empirical results are strong.

Initial concerns centered on (1) clarity and missing technical details (problem formulation, simulator, RL algorithm, and GNN architecture); (2) the necessity and benefit of the dual-policy factorization versus a single joint policy; (3) scalability issues (to large scale, e.g., >8 GPUs); (4) impracticality of Stage III; and (5) missing explanation or studies for convergence or sample complexity analysis of the dual-policy setup.

The rebuttal and revision directly address most of these points with additional experiments and clarifications. Reviewers kPCd, hjpH, qMoz, and navF explicitly state that their concerns were addressed. Given the strong empirical performance, the additional ablations, and the reviewers’ acknowledgment that the issues were resolved, I lean toward acceptance as an empirically strong systems paper, despite remaining limitations in theory and very large-scale evaluation.

**Reviewer Concerns:**

Below I summarize, which concerns were resolved and which remain only partially addressed.
1.  Clarity issues and missing technical details (problem formulation, simulator, RL algorithm, and GNN architecture);  These issues are well addressed.  The authors expanded the abstract to include quantitative results, clarified the definition and role of the simulator, retitled Sections 2–3 to better separate background from the problem/RL formulation, and moved key technical details from the appendix into the main text.
(2) the necessity and benefit of the dual-policy factorization versus a single joint policy; (3) scalability issues (to large scale, e.g., >8 GPUs); (4) impracticality of Stage III; and (5) missing explanation or studies for convergence or sample complexity analysis of the dual-policy setup.
2. The necessity and benefit of the dual-policy factorization versus a single joint policy. During the rebuttal, the authors provide extensive ablation studies and justifications for these issues.
3. Scalability issues (to large scale, e.g., >8 GPUs).   Not fully addressed empirically. The authors do not add large-scale experiments; instead, they argue that their setting is representative of practical inference workloads (typically only a few dozen GPUs) and that large-scale training is dominated by many repeated small subgraphs where the same learned policy could be reused. They leave full large-cluster evaluation to future work.

4. Impracticality of Stage III. Partially addressed. The authors clarify that Stage III is not intended to train from scratch (the 0k–0k–8k setting is only an ablation), but rather to perform sim-to-real adaptation starting from a well-trained policy, which makes it more practical than originally perceived.

5. Theoretical guarantees and convergence / sample complexity.  The authors acknowledge that formal analysis is challenging and leave this to future work. Given the empirical, systems-oriented nature of the paper, this is acceptable but still a notable limitation.

In summary, this is a somewhat borderline paper, but the main technical and presentation concerns have been substantively addressed. The primary remaining weaknesses are the lack of formal theory and the absence of very-large-scale experiments, which the authors acknowledge and partially justify. Given the strong empirical results and the reviewers’ post-rebuttal positions, I still regard this as a strong empirical systems paper and lean toward acceptance.

**Reviewer Scores:**

Reviewer qMoz (2 --> improved to 4 or maybe 6), Reviewer qMoz (2 --> improved to 4 or maybe 6), Reviewer kPCd (8 --> 8),  Reviewer hjpH (4 --> maybe slightly improved to 6), Reviewer navF (4 --> maybe slightly improved to 6)

---

### Decision · Program_Chairs · 2026-01-26

Accept (Poster)